# Δ133p53α and Δ160p53α isoforms of the tumor suppressor protein p53 exert dominant-negative effect primarily by co-aggregation

**Liuqun Zhao[1†], Tanel Punga[2†], Suparna Sanyal[1]\***

[1]Department of Cell and Molecular Biology, Uppsala University, Biomedical Center, Uppsala, Sweden; [2]Department of Medical Biochemistry and Microbiology, Uppsala University, Biomedical Center, Uppsala, Sweden

## eLife Assessment

This **important** study investigates the molecular mechanisms by which the p53 isoforms Δ133p53α and Δ160p53α exert dominant-negative effects on full-length p53 (FLp53). Through a combination of chromatin immunoprecipitation, transcriptional reporter assays, subcellular localization analyses, and protein aggregation experiments, the authors provide **solid** evidence that these N-terminally truncated isoforms promote co-aggregation with FLp53, disrupting its transcriptional activity and cellular distribution. The revised manuscript successfully addresses prior reviewer concerns, and the findings are well supported by the experimental data.
[Editors' note: this paper was reviewed by Review Commons.]

**\*For correspondence:**
suparna.sanyal@icm.uu.se

[†]These authors contributed equally to this work

**Competing interest:** The authors declare that no competing interests exist.

## Abstract

p53 is a tumor suppressor protein with multiple isoforms with shared or specific functions. However, two of its isoforms, Δ133p53α and Δ160p53α, with large N-terminal deletions, can cause cancer. These isoforms exert a dominant-negative effect on full-length p53 (FLp53), although the precise molecular mechanisms are unknown. Here, we investigate the mechanisms of action of Δ133p53α and Δ160p53α isoforms using chromatin immunoprecipitation, luciferase expression, subcellular fractionation, immunofluorescence assays, and apoptotic caspase activity assay. Our study elucidates that these DNA-binding deficient p53 isoforms form hetero-tetrameric complexes with FLp53 and disrupt FLp53's DNA binding and transcriptional activities when present in a higher proportion than FLp53 in the tetramer. However, these structurally unstable isoforms promote vigorous protein aggregation involving FLp53, disrupting its structure and sequestering it in the cytoplasmic and nuclear aggregates, thereby limiting its availability to function as a transcription activator protein. Thus, co-aggregation of Δ133p53α and Δ160p53α with FLp53, rather than hetero-tetramerization, is likely the primary factor contributing to their dominant-negative effect. Modulating the stability and aggregation of p53 isoforms could be a novel strategy for cancer therapy.

## Introduction

The tumor suppressor protein p53 is an essential transcription activator protein modulating the expression of multiple target genes. By regulating various cellular processes such as cell cycle arrest, apoptosis, senescence, DNA repair, and angiogenesis, p53 prevents cancer development (*Levine and Oren, 2009*; *Vogelstein et al., 2000*). The intricate p53 pathway is stimulated by various stress stimuli, including DNA damage and oncogene activation, which lead to the stabilization and activation

**eLife digest** Cancer is a deadly disease that continues to pose a major challenge to global health-care systems. It arises when some cells acquire mutations that allow them to grow and divide uncontrollably. A key player for preventing this uncontrolled cell growth is the protein p53, which binds to DNA and regulates the activities of genes that determine whether a cell should grow, divide, or die.

Mutations in the DNA-binding domain of p53 can hinder its ability to regulate gene activity, thereby contributing to cancer. Additionally, shorter versions of p53 – known as p53 isoforms – can accumulate in the cell and coexist with the normal full-length protein. These isoforms sometimes lack critical components of the protein, whilst retaining other properties. For example, two p53 isoforms – called Δ133p53α and Δ160p53α – are missing parts of the DNA-binding domain but can still bind to full-length p53 protein. Here, Zhao, Punga and Sanyal used biochemical and cell biology approaches to investigate how the Δ133p53α and Δ160p53α isoforms affect the activity of full-length p53.

The team found that both isoforms can bind to full-length p53 protein and form tetramers, large molecules made up of four p53 sub-units. In addition, they found that the Δ133p53α and Δ160p53α isoforms can form aggregates with full-length p53. As a result, full-length p53 protein gets sequestered within these aggregates, which prevents them from binding to DNA and carrying out their role in gene expression and cell-death regulation.

These findings open up new avenues for cancer research, particularly in cancer patients who have truncated isoforms of the p53 protein. Developing new therapies that prevent shorter and normal full-length p53 protein from forming irregular aggregates may have the potential to prevent, or slow, cancer progression in these patients.

of the p53 protein (*Pflaum et al., 2014*). Stabilized p53 acts as a transcription activator protein as it can directly bind to its response elements (REs) and recruit different transcription co-activator proteins at various gene promoters (*Sullivan et al., 2018*). One of the best-known p53 targets is the p21 (also known as CIP1, WAF1) gene promoter, which contains two p53REs and is efficiently activated by the p53 protein (*el-Deiry et al., 1993*; *Macleod et al., 1995*). The encoded p21 protein blocks cell cycle progression and thereby executes tumor suppressor activity of the p53 protein (*Indeglia and Murphy, 2024*; *Shamloo and Usluer, 2019*).

The TP53 gene produces at least 12 distinct p53 isoforms: p53 (α, β, γ), Δ40p53 (α, β, γ), Δ133p53 (α, β, γ), and Δ160p53 (α, β, γ). It has been shown that the p53 isoforms contribute to the regulation of the p53 pathway by regulating the expression of p53 target genes (*Joruiz and Bourdon, 2016*; *Khoury and Bourdon, 2011*; *Zhao and Sanyal, 2022*). Notably, p53 isoforms exhibit both overlapping and distinct functions compared to canonical p53 (*Mehta et al., 2021*; *Pal et al., 2023*; *Zhao and Sanyal, 2022*) and have been reported to promote tumor progression in various cancers (*Vieler and Sanyal, 2018*; *Zhao and Sanyal, 2022*). The p53 isoforms are generated through alternative promoter usage, alternative splicing, and alternative transcription starting site usage (*Joruiz and Bourdon, 2016*; *Sharathchandra et al., 2014*). These isoforms differ from the full-length (FL) p53 (canonical p53, p53α) by lacking variable segments of the N-terminus (N-terminal isoforms: Δ40, Δ133, and Δ160) and possessing alternative C-terminus (C-terminal isoforms: α, β, and γ). The FLp53 protein consists of four primary functional domains: transactivation domains (TAD-1 and TAD-2), DNA-binding core domain (DBD), nuclear localization sequence (NLS), and the oligomerization domain (OD) (*Khoury and Bourdon, 2010*). Native FLp53 typically functions as a tetrameric protein, where the individual monomers bind cooperatively to DNA (*Jeffrey et al., 1995*; *Weinberg et al., 2004*). The C-terminal truncated isoforms (β/γ isoforms) lacking the OD have lower transactivation activity because of reduced tetramerization efficiency (*Jänicke et al., 2009*; *Kim et al., 2012*; *Wang et al., 1994*; *Wang et al., 1995*). In contrast, the N-terminal truncated isoforms (α isoforms), which retain the entire C-terminal OD, can form various complexes with FLp53 and exert dominant-negative effects on p53's transcriptional activity (*Bourdon et al., 2005*; *Fujita et al., 2009*; *Horikawa et al., 2017*). This study focuses on two large N-terminal deleted p53 isoforms: Δ133p53α and Δ160p53α, the latter has been largely unexplored. For convenience and simplicity, we have written Δ133p53 and Δ160p53 to represent the α isoforms (Δ133p53α and Δ160p53α) throughout this manuscript. The Δ133p53 protein lacks both TADs and a part of the first conserved cysteine box of the DBD, whereas

the Δ160p53 protein lacks both TADs and the entire first conserved cysteine box of the DBD (*Joruiz and Bourdon, 2016*; *Pavletich et al., 1993*).

The Δ133p53 isoform exhibits complex biological functions, with both oncogenic and non-oncogenic potentials. Recent studies demonstrate the non-oncogenic yet context-dependent role of the Δ133p53 isoform in cancer development. Δ133p53 expression has been reported to correlate with improved survival in patients with TP53 mutations (*Bischof et al., 2019*; *Hofstetter et al., 2011*), where it promotes cell survival in a non-oncogenic manner (*Gong et al., 2015*; *Gong et al., 2016b*), especially under low oxidative stress (*Gong et al., 2016c*). Alternatively, other recent evidence emphasizes the notable oncogenic functions of Δ133p53 as it can inhibit p53-dependent apoptosis by directly interacting with the FLp53 (*Aoubala et al., 2011*; *Bourdon et al., 2005*). The oncogenic function of the newly identified Δ160p53 isoform is less known, although it is associated with p53 mutation-driven tumorigenesis (*Candeias et al., 2016*) and in melanoma cells' aggressiveness (*Tadijan et al., 2021*). Whether or not the Δ160p53 isoform also impedes FLp53 function in a similar way as Δ133p53 is an open question. However, these p53 isoforms can certainly compromise p53-mediated tumor suppression by interfering with FLp53 binding to target genes such as p21 and miR-34a (*Fujita et al., 2009*; *Horikawa et al., 2017*) by dominant-negative effect, and the exact mechanism is not known.

Protein aggregation has become a central focus of modern biology research and has documented implications in various diseases, including cancer (*Farmer et al., 2020*; *Forget et al., 2013*; *Petronilho et al., 2024*). Protein aggregates can be of different types, ranging from amorphous aggregates to highly structured amyloid or fibrillar aggregates, each with different physiological implications. In the case of p53, whether protein aggregation, and, in particular, co-aggregation with large N-terminal deletion isoforms, plays a mechanistic role in its inactivation is yet underexplored. Interestingly, the Δ133p53β isoform has been shown to aggregate in several human cancer cell lines (*Arsic et al., 2021*). Additionally, the Δ40p53α isoform exhibits a high aggregation tendency in endometrial cancer cells (*Melo Dos Santos et al., 2019*). Although no direct evidence exists for Δ160p53 yet, these findings imply that p53 isoform aggregation may play a major role in their mechanisms of actions.

The precise molecular mechanisms involved in the biological function of the Δ133p53 and the Δ160p53 proteins, especially how they exert dominant-negative effects on FLp53, remain poorly understood. In this study, we have analyzed the DNA-binding, transcriptional activity, and oligomerization capacity of the human FLp53, Δ133p53, and Δ160p53 proteins. Our data show that the Δ133p53 and Δ160p53 proteins are deficient in binding to p53-regulated promoters and inducing expression of the p53 target genes. Moreover, the Δ133p53 and Δ160p53 isoforms can form hetero-oligomers with FLp53 and inhibit FLp53 function in a concentration-dependent manner. Two potential mechanisms for their dominant-negative effects have been suggested: (i) the formation of hetero-tetramers with FLp53, inhibiting its DNA-binding and transcriptional activity, and (ii) the aggregation property of the isoforms leading to co-aggregation of the FLp53 protein. Interestingly, while hetero-tetramerization partially attenuates p53 activity, the co-aggregation of FLp53 with Δ133p53 and Δ160p53 emerges as the predominant factor contributing to the dominant-negative effect.

## Results

### Δ133p53 and Δ160p53 proteins exhibit structural destabilization, exposure of hydrophobic surfaces, and high propensity to aggregation

The DBD, also known as the core domain of the FLp53 protein (human p53, residues 94–292), has a structure with an aggregation-prone sequence (residues 251–257) (*Xu et al., 2011*; *Figure 1A*). The core domain contains a hydrophobic core of β-sandwich, which consists of two antiparallel β-sheets with four strands (S1, S3, S8, and S5) and five strands (S6, S7, S4, S9, and S10), respectively (*Cho et al., 1994*; *Figure 1—figure supplement 1A*). The DNA-binding surface is constituted by two large loops (L2 and L3) and a loop-sheet-helix (LSH) motif, which comprises loop L1, a β-sheet formed by the β-hairpin S2–S2', the C-terminal residues of strand S10, and the α-helix H2 (*Joerger et al., 2004*; *Figure 1—figure supplement 1A*). Compared to FLp53, the N-terminal deletion isoforms Δ133p53 and Δ160p53 lack the entire TADs (*Figure 1A*), which are crucial for recruiting transcription co-activator proteins (*Raj and Attardi, 2017*). In addition, the Δ133p53 protein is missing the β-strand S1 in the β-sandwich and a part of the LSH motif (L1, S2, and the N-terminal residue 132 of the strand S2') in the DBD (*Figure 1—figure supplement 1A*). The Δ160p53 protein, on the other hand, lacks a part

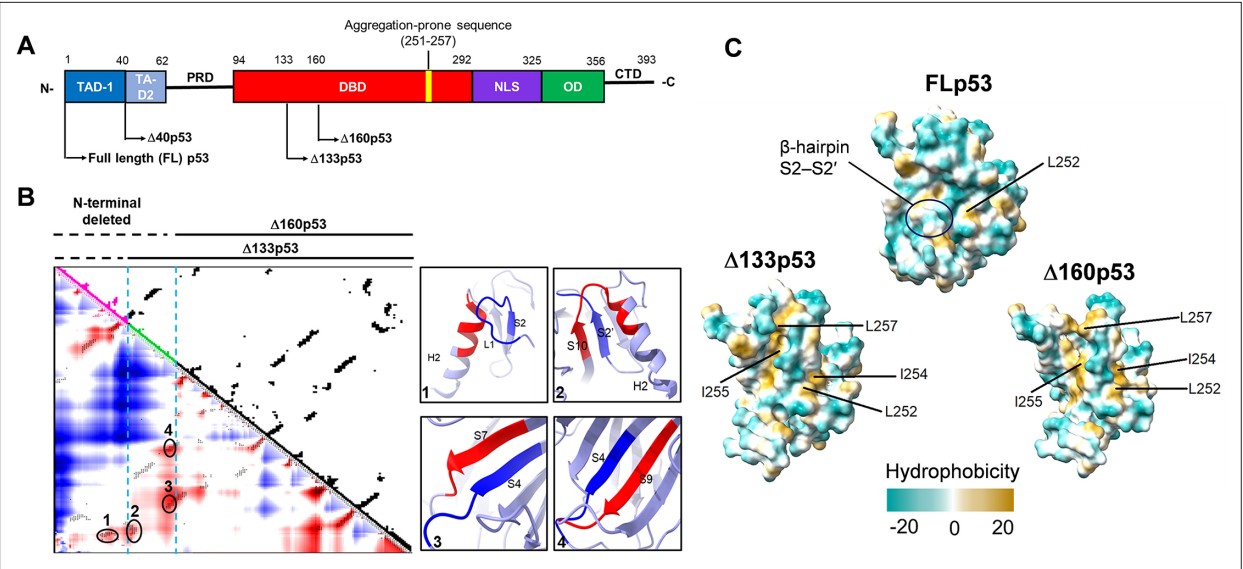

**Figure 1.** The Δ133p53 and Δ160p53 proteins exhibit structural destabilization. (**A**) Schematic domain structures of full-length (FL) p53 and its isoforms. TAD, transactivation domain; PRD, proline-rich domain; DBD, DNA-binding domain; NLS, nuclear localization sequence; OD, oligomerization domain; CTD, C-terminal domain. Δ133p53 lacks the entire TAD and approximately 20% of the DBD. Δ160p53 lacks the entire TAD and approximately 33% of the DBD. An aggregation region ILTIITL (residues 251–257) is indicated in the DBD domain. (**B**) Contact density map (C-α 8 Å) of the core domain of FLp53 (PDB ID: 3KMD) using CWView (*Vehlow et al., 2011*). The residues 94–132 and 133–159 were colored magenta and green, respectively. Higher densities are shown in red, and low densities are shown in blue. Among these contacts, four higher-density contacts were selected and shown in oval circles on the map. The corresponding secondary structures of the contacts between the deleted region (blue) and the region (red) in the loop-sheet-helix (LSH) motif or the β-sandwich are shown on the right. (**C**) Comparison analysis of surface hydrophobicity of the core domain in FLp53 (PDB ID: 3KMD) and its isoforms Δ133p53 and Δ160p53 by ChimeraX (*Goddard et al., 2018*; *Meng et al., 2023*). The exposed hydrophobic residues in the aggregation region ILTIITL (residues 251–257) are labeled. The color scale of hydrophobicity is indicated with the maximum and minimum values in dark cyan and dark goldenrod, respectively. The β-hairpin S2–S2′ was shown in an oval circle.

The online version of this article includes the following source data and figure supplement(s) for figure 1:

**Figure supplement 1.** Core domain structures of full-length (FL) p53 and its isoforms Δ133p53 and Δ160p53.

**Figure supplement 2.** Aggregation propensity of FLp53 and its Δ133p53 and Δ160p53 isoforms by sedimentation analysis in SDS-PAGE.

**Figure supplement 2—source data 1.** Original SDS-PAGE gel images corresponding to *Figure 1—figure supplement 2*, labeled.

**Figure supplement 2—source data 2.** Original SDS-PAGE gel images corresponding to *Figure 1—figure supplement 2*.

of the β-sandwich (β-strands S1, S3, and N-terminal residues 156–159 of S4) and a part of the LSH motif (L1 and β-hairpin S2–S2′) (*Figure 1—figure supplement 1A*). Therefore, the N-terminal deletion affects the structural integrity of both the β-sandwich and the LSH motif in the core domain.

To assess the structural impact of the N-terminal deletions in FLp53, we analyzed the contact information derived from the core domain of FLp53 (PDB ID: 3KMD) structure (*Chen et al., 2010*). We observed that both Δ133p53 and Δ160p53 lack several crucial contacts between the N- and C-terminal regions in FLp53 (*Figure 1B*). For Δ133p53, the contact between the L1 and S2 segments in the N-terminal region and the α-helix H2 in the C-terminal region within the LSH motif is disrupted (*Figure 1B*, insert 1), suggesting a potential impairment of the stability of DNA-binding interface. For Δ160p53, in addition to the missing contact between L1, S2, and H2, there are also no contacts between the S2′ segment in the N-terminal region and the S10 and H2 segments in the C-terminal region within the LSH motif (*Figure 1B*, insert 2), as well as between S4 and S7 or S9 in the β-sandwich core domain (*Figure 1B*, inserts 3 and 4). These contacts are crucial for maintaining the structural integrity and folding of the core domain. Therefore, both Δ133p53 and Δ160p53 will likely have a partially or fully destabilized core domain that may impair their DNA-binding ability.

We predicted the monomer structure of Δ133p53 and Δ160p53 by AlphaFold2 (*Mirdita et al., 2022*; *Figure 1—figure supplement 1B*). Hydrophobic surface analysis revealed that Δ133p53 and Δ160p53 exposed four highly hydrophobic residues (L252, I254, I255, and L257) within the aggregation-prone sequence, while FLp53 exposes only one residue (L252) (*Figure 1C*). This increased hydrophobicity may induce the aggregation of Δ133p53 and Δ160p53. Moreover, Δ133p53 and Δ160p53 have a

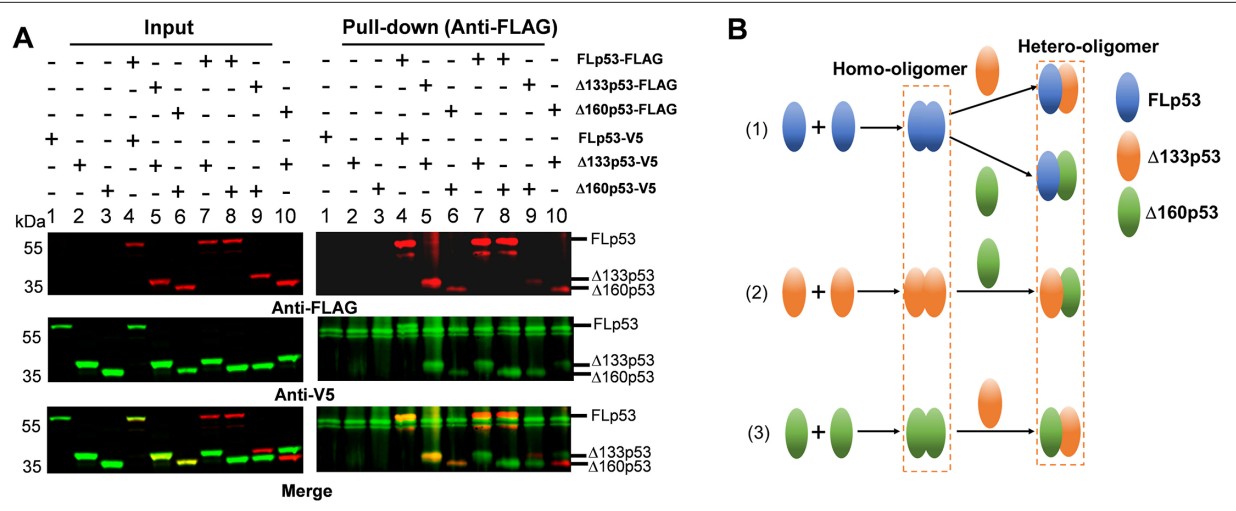

**Figure 2.** Complex formation among FLp53 and its isoforms Δ133p53 and Δ160p53. (**A**) Co-immunoprecipitation of FLp53-FLAG, Δ133p53-FLAG, and Δ160p53-FLAG along with the FLp53-V5, Δ133p53-V5, and Δ160p53-V5 proteins in H1299 cells. The complexes were isolated with anti-FLAG antibody-coupled magnetic beads. Western blot analysis was done with the anti-FLAG and anti-V5 antibodies. The merged image corresponds to the overlay between the FLAG and V5 antibody signals. Input corresponds to 5% of the whole-cell lysate. (**B**) A summary of FLp53, Δ133p53, and Δ160p53 oligomerization based on data shown in panel A.

The online version of this article includes the following source data and figure supplement(s) for figure 2:

**Source data 1.** Original membranes corresponding to *Figure 2A*, labeled.

**Source data 2.** Original membranes corresponding to *Figure 2A*.

**Figure supplement 1.** Hetero-oligomers formation between full-length p53 (FLp53) and its isoforms Δ133p53 or Δ160p53.

**Figure supplement 1—source data 1.** Original membranes corresponding to *Figure 2—figure supplement 1*, labeled.

**Figure supplement 1—source data 2.** Original membranes corresponding to *Figure 2—figure supplement 1*.

partial or complete structural loss of the β-hairpin S2–S2′ (*Figure 1—figure supplement 1A*), which restricts access to the hydrophobic core (*Cañadillas et al., 2006*). This structural alteration results in an enhanced exposure of the hydrophobic core of the β-sandwich domain compared to FLp53 (*Figure 1C*). This exposure potentially facilitates the propensity for protein aggregation, as suggested by previous studies (*Cañadillas et al., 2006*; *Olotu and Soliman, 2018*) and our in vitro aggregation assay with purified FLp53, Δ133p53, and Δ160p53 proteins (*Figure 1—figure supplement 2*). In summary, these findings suggest that the large N-terminal deletion in Δ133p53 and Δ160p53 would compromise the structural stability of FLp53 and induce aggregation, with Δ160p53 being more severely affected among the two.

## Δ133p53 and Δ160p53 form complexes with FLp53

To investigate the oligomerization properties of Δ133p53 and Δ160p53, which retain the OD known to facilitate p53 tetramerization (*Clore et al., 1995*), we co-expressed the FLAG-tagged p53 proteins as the baits along with the V5-tagged FLp53, Δ133p53, and Δ160p53 proteins as preys. The V5-tagged FLp53, Δ133p53, and Δ160p53 proteins specifically interacted with their corresponding FLAG-tagged proteins (*Figure 2A*, lanes 1–6), demonstrating their capacity to form homo-oligomers (*Figure 2B*). In addition, interactions between FLp53-FLAG/Δ133p53-V5, FLp53-FLAG/Δ160p53-V5, Δ133p53-FLAG/Δ160p53-V5, and Δ160p53-FLAG/Δ133p53-V5 were found (*Figure 2A*, lanes 7–10), suggesting that they can also form hetero-oligomers (*Figure 2B*). These findings are consistent with previous studies, indicating that Δ40p53 can hetero-oligomerize with FLp53 (*Courtois et al., 2002*; *Hafsi et al., 2013*). To rule out the possibility that the observed interactions between FLp53 and its isoforms Δ133p53 and Δ160p53 were artifacts caused by the FLAG and V5 antibody epitope tags, we co-expressed FLAG-tagged FLp53 with untagged Δ133p53 and Δ160p53. Immunoprecipitation assays demonstrated that FLAG-tagged FLp53 could still interact with the untagged Δ133p53 and Δ160p53 isoforms (*Figure 2—figure supplement 1*, lanes 3 and 4), confirming the formation of hetero-oligomers between FLp53

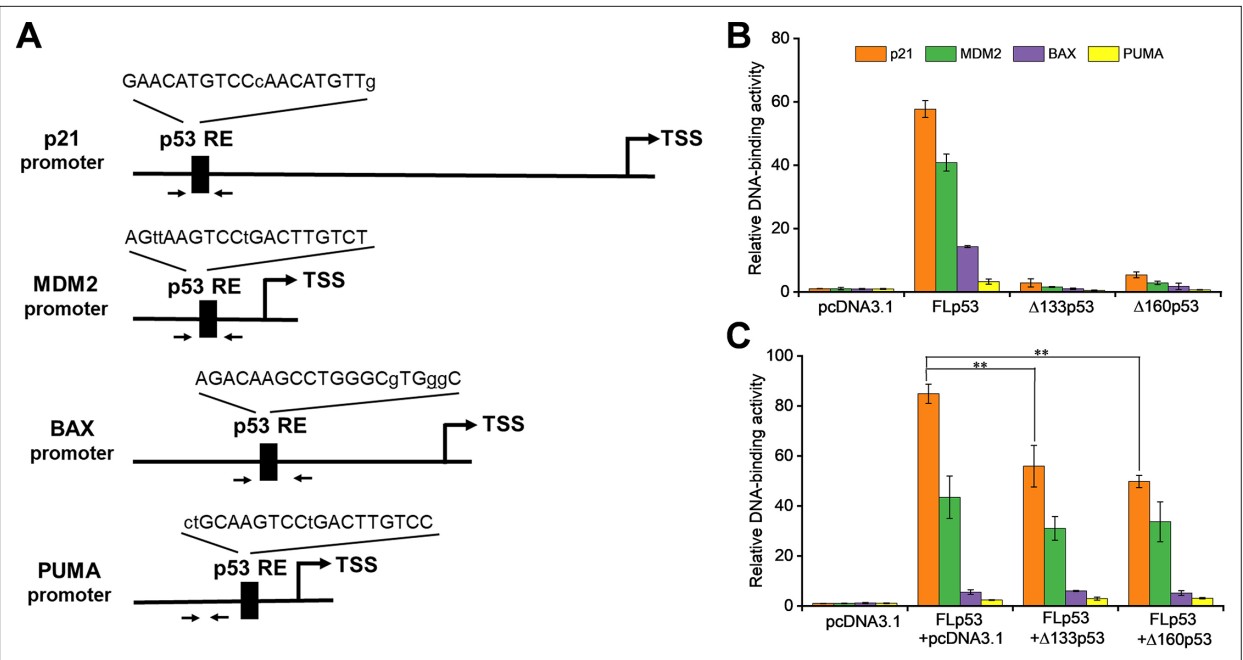

**Figure 3.** Δ133p53 and Δ160p53 negatively influence FLp53's DNA-binding activity. (**A**) Schematic representation of the promoters of p53 target genes (p21, MDM2, BAX, and PUMA). An arrow denotes the transcription start site (TSS). The DNA sequences of p53 response elements (REs) are shown, with uppercase and lowercase letters indicating matched and mismatched bases, respectively, in relation to the canonical p53 RE sequence (RRRCWWGYYY). Paired arrows highlight the regions subjected to quantitative PCR (qPCR) amplification. Specifically, the distal (5') p53 RE within the p21 gene promoter was analyzed. (**B**) Relative DNA binding of the FLp53-FLAG, Δ133p53-FLAG, and Δ160p53-FLAG proteins to p53 target genes (p21, MDM2, BAX, and PUMA) in H1299 cells. (**C**) Relative DNA binding of the FLp53-FLAG protein to the p53 target gene promoters in the presence of the V5-tagged protein Δ133p53 or Δ160p53 at a 1:1 ratio. Chromatin immunoprecipitation (ChIP)-qPCR assay data are shown as relative enrichment of promoter sequences of the target genes after normalization to the control, pcDNA3.1 plasmid transfected cells. Data are represented as the mean of technical triplicates ± standard deviation (SD), **p<0.01 (Student's t-test).

The online version of this article includes the following source data for figure 3:

**Source data 1.** Chromatin immunoprecipitation (ChIP)-quantitative PCR (qPCR) assay data corresponding to *Figure 3B and C*.

## Δ133p53 and Δ160p53 negatively influence FLp53's DNA-binding activity

Previous studies have shown that the p53 lacking the TADs and the PRD (ΔNp53DBD, residues 94–393) retain full DNA-binding capacity as the FLp53 (*Emamzadah et al., 2014*; *Petty et al., 2011*). In contrast, the Δ133p53 (residues 133–393) and Δ160p53 (residues 160–393) isoforms additionally lack part of the DBD (residues 94–292). This deficiency raises questions about the impact of partial DBD deletion on the DNA-binding activity of p53. To address this question, we performed a chromatin immunoprecipitation (ChIP) assay to measure the DNA-binding activity of FLp53 and its isoforms Δ133p53 and Δ160p53 to several known p53-targeted promoters controlling proteins regulating cycle arrest (p21), protein stability (MDM2), and apoptosis induction (BAX and PUMA) (*Figure 3A*). Previous studies have classified p53 DNA-binding affinities. Accordingly, the p21 promoter has the highest affinity, the MDM2 promoter has moderate affinity, whereas the BAX and PUMA promoters show relatively low p53-binding affinity (*Park et al., 2012*; *Purvis et al., 2012*; *Weinberg et al., 2005*). Consistent with these findings, our results revealed distinct binding affinities of the FLp53 protein to these target gene promoters (*Figure 3B*). Notably, the Δ133p53 and Δ160p53 isoforms demonstrated a significant loss of DNA-binding activity at these promoters (*Figure 3B*).

We subsequently hypothesized that Δ133p53 and Δ160p53 might compromise the DNA-binding ability of FLp53 through hetero-oligomer formation (*Figure 2*). To study this, we co-transfected FLp53 along with either Δ133p53 or Δ160p53 expressing plasmid and measured FLp53's binding to the

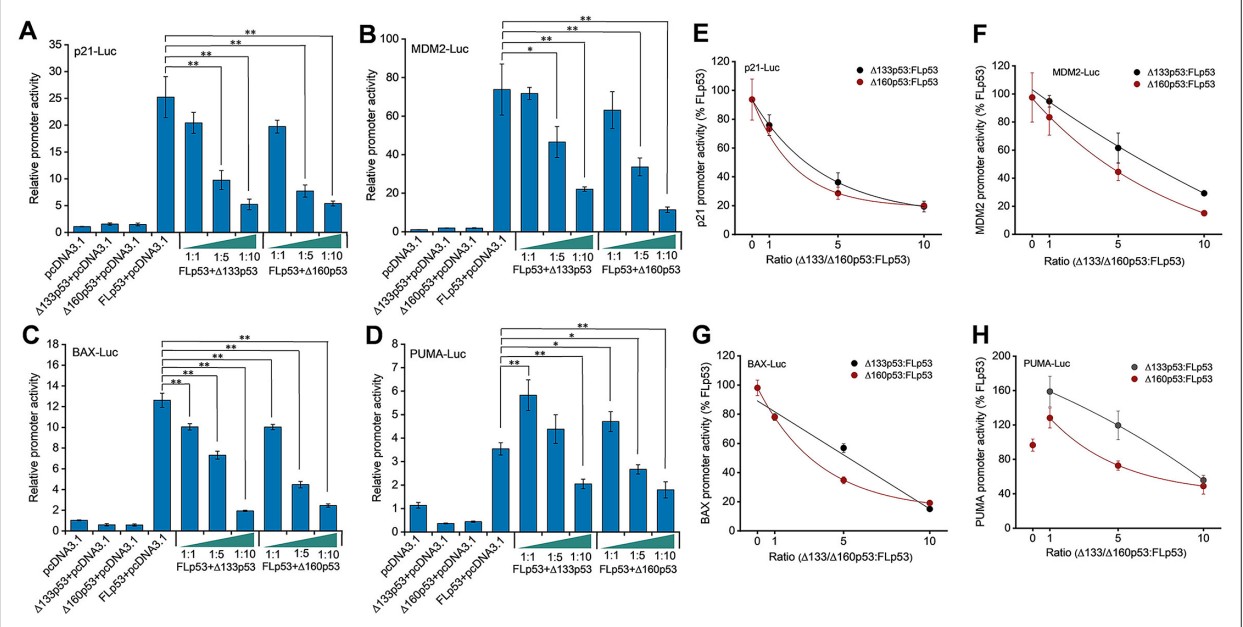

**Figure 4.** Impact of Δ133p53 and Δ160p53 on FLp53-activated transactivation. (**A–D**) H1299 cells were transfected with luciferase reporter plasmids driven by p21 (**A**), MDM2 (**B**), BAX (**C**), and PUMA (**D**) promoters, along with vectors expressing FLp53, Δ133p53, or Δ160p53. To evaluate the influence of the isoforms on FLp53's transactivation capability, co-expression was performed at ratios of 1:1, 1:5, and 1:10 relative to FLp53. Basal promoter activity was determined by transfecting cells with the empty vector, pcDNA3.1. Relative promoter activity is shown after normalization to the pcDNA3.1-treated sample activity. Data represent mean values ± standard deviation (SD) (n=3). *p<0.05; **p<0.01 (Student's t-test). (**E–H**) The transcriptional activity of FLp53 on the p21 (**E**), MDM2 (**F**), BAX (**G**), and PUMA (**H**) promoters was inhibited by Δ133p53 and Δ160p53. Inhibition curve fitting was performed using the exponential function ExpDec1 in Origin 2018 software.

The online version of this article includes the following source data and figure supplement(s) for figure 4:

**Source data 1.** Luciferase reporter assay data corresponding to *Figure 4A–D*.

**Source data 2.** Data corresponding to *Figure 4E–H*, showing the inhibition of FLp53's transcriptional activity by Δ133p53 and Δ160p53.

**Figure supplement 1.** Predictive inhibitory capacity of Δ133p53 and Δ160p53 isoforms in hetero- or homo-tetrameric complexes.

aforementioned promoters. The presence of Δ133p53 or Δ160p53 significantly reduced the binding of FLp53 to the p21 promoter, which has the highest affinity for FLp53, while having minimal impact on its interaction with the other promoters (*Figure 3C*). These findings suggest that Δ133p53 and Δ160p53 isoforms may selectively impair the ability of FLp53 to bind to high-affinity target promoters.

## Δ133p53 and Δ160p53 impede FLp53-mediated transcription activation

The p53-mediated transcription activation depends on its binding ability to specific p53REs at the target gene promoters (*Beckerman and Prives, 2010*; *Menendez et al., 2013*). To evaluate the influence of the FLp53 protein interaction with its isoforms, Δ133p53 and Δ160p53, on transcription, we employed a luciferase reporter assay with the p21, MDM2, BAX, and PUMA promoters fused to a firefly luciferase reporter gene. As expected, all four promoters were responsive to FLp53-mediated transcriptional activation (*Figure 4A–D*). Notably, FLp53 activation of the PUMA promoter was relatively low, possibly due to the lower binding affinity of FLp53 to the PUMA promoter (*Figure 3B*). In contrast, neither Δ133p53 nor Δ160p53 activated any tested promoters (*Figure 4A–D*), indicating a lack of functional capability to stimulate transcription from these promoters.

In order to assess how Δ133p53 and Δ160p53 influence FLp53-mediated transcription on different p53-responsive promoters, we co-expressed FLp53 along with increasing amounts of Δ133p53 or Δ160p53 in H1299 cells. We found that co-expression of FLp53 with Δ133p53 or Δ160p53 at a 1:1 ratio did not significantly alter the activation of the p21 and MDM2 promoters compared to FLp53 alone (*Figure 4A and B*). This result suggests that at an equal protein expression ratio, these isoforms did not exert a dominant-negative effect on the FLp53 at these promoters. However, a strong reduction in

BAX promoter activation was evident under the same experimental condition (*Figure 4C*), suggesting that the isoforms can disrupt FLp53 function on the BAX promoter. Interestingly, both isoforms enhanced the activation of the PUMA promoter when added with FLp53 at a 1:1 ratio (*Figure 4D*).

The elevated Δ133p53 protein modulates p53 target genes such as miR-34a and p21, facilitating cancer development (*Fragou et al., 2017*; *Fujita et al., 2009*). To mimic conditions where isoforms are upregulated relative to FLp53, we increased the ratios to 1:5 and 1:10. Adjusting the ratio of FLp53 to isoforms to 1:5 revealed that higher isoform expression levels significantly impaired FLp53-mediated activation of the p21, MDM2, and BAX promoters (*Figure 4A–C*). Interestingly, Δ133p53 at a 1:5 ratio had almost no impact on PUMA promoter activity, whereas Δ160p53 reduced it compared to FLp53 alone (*Figure 4D*). These results indicate that higher expression ratios of Δ133p53 and Δ160p53 relative to FLp53 might be required to exert dominant-negative effects. To test this hypothesis, we escalated the ratio of FLp53 to isoforms to 1:10. As expected, the activity of all four promoters decreased significantly at this ratio (*Figure 4A–D*). Notably, Δ160p53 showed a more potent inhibitory effect than Δ133p53 at the 1:5 ratio on all promoters except for the p21 promoter, where their impacts were similar (*Figure 4E–H*). However, at the 1:10 ratio, Δ133p53 and Δ160p53 had similar effects on all transactivation except for the MDM2 promoter (*Figure 4E–H*). This differential impact underscores the variability in how p53 isoforms influence FLp53 function across different promoters and expression ratios. Collectively, these findings demonstrate that isoforms with impaired DNA-binding capability can indeed exert dominant-negative effects on the function of FLp53, with the magnitude of this impact being promoter-specific and dependent on the relative expression levels of the proteins.

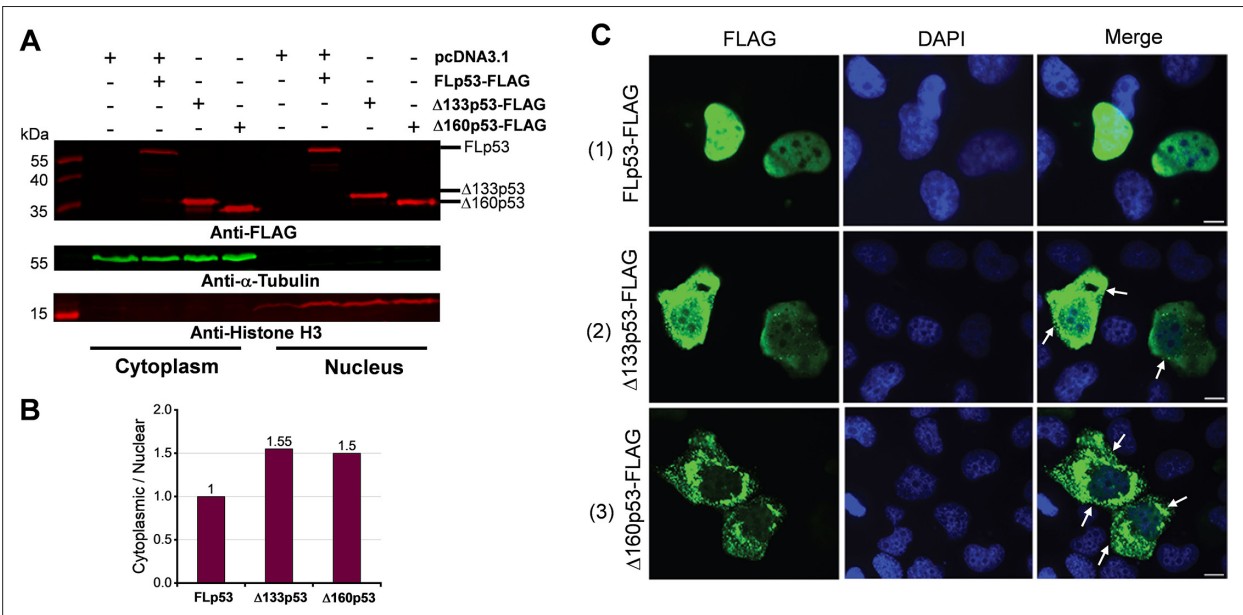

**Figure 5.** Subcellular distribution of the FLp53, Δ133p53, and Δ160p53 proteins. (**A**) Western blot analysis of subcellular fractionation of H1299 cells transfected with the FLAG-tagged FLp53, Δ133p53, or Δ160p53 proteins. Representative immunoblots indicate the presence of these proteins in different cellular fractions. α-Tubulin (cytoplasmic marker) and Histone H3 (nuclear marker) were used to ensure fraction integrity. (**B**) Quantitative bar graph summarizing the cytoplasm-to-nucleus ratio of protein levels. FLAG-tagged protein levels were normalized to their respective fractionation markers: α-Tubulin for the cytoplasmic fraction and Histone H3 for the nuclear fraction. The relative ratios of Δ133p53 and Δ160p53 protein levels are displayed, with the FLp53 cytoplasm-to-nucleus ratio set as 1. (**C**) Immunofluorescence analysis of H1299 cells expressing FLAG-tagged FLp53, Δ133p53, and Δ160p53. Cell nuclei are visualized with DAPI (blue). The punctate spots (white arrows) indicating protein aggregates are marked. Scale bar, 50 μm.

The online version of this article includes the following source data and figure supplement(s) for figure 5:

**Source data 1.** Original membranes corresponding to *Figure 5A*, labeled.

**Source data 2.** Data corresponding to *Figure 5B*, showing the cytoplasm-to-nucleus ratio of protein levels.

**Source data 3.** Original immunofluorescence images corresponding to *Figure 5C*.

**Source data 4.** Original membranes corresponding to *Figure 5A*.

**Figure supplement 1.** Immunofluorescence analysis of full-length p53 (FLp53) localization in H1299 cells.

**Figure supplement 1—source data 1.** Original immunofluorescence images corresponding to *Figure 5—figure supplement 1*.

## The Δ133p53 and Δ160p53 proteins affect the subcellular localization and aggregation of FLp53

p53 exerts its transcriptional activity in the nucleus, and its appropriate subcellular localization is crucial for regulating p53 functions (*Stommel et al., 1999*). Immunoblotting analysis with isolated nuclear and cytoplasmic fractions revealed distinct subcellular localization patterns for FLp53 and its isoforms Δ133p53 and Δ160p53 (*Figure 5A and B*). The cytoplasm-to-nucleus ratio of Δ133p53 and Δ160p53 was approximately 1.5-fold higher than that of FLp53 (*Figure 5B*). These results suggest that Δ133p53 and Δ160p53 exhibit a greater tendency for cytoplasmic localization.

Immunofluorescence analysis of the cells expressing FLp53 and the two isoforms confirmed the above-observed protein localization pattern (*Figure 5C*). The FLp53 protein was predominantly localized in the nucleus, with minimal to no fluorescence signal detected in the cytoplasm (*Figure 5C*, panel 1 and *Figure 5—figure supplement 1*). In contrast, the Δ133p53 and Δ160p53 proteins exhibited reduced nuclear staining and enhanced cytoplasmic staining relative to FLp53, with Δ160p53 showing more pronounced cytoplasmic staining (*Figure 5C*). Notably, Δ133p53 and Δ160p53 demonstrated a heterogeneous distribution in the nucleus and cytoplasm (*Figure 5C*, panels 2 and 3). Furthermore, both Δ133p53 and Δ160p53 exhibited predominantly cytoplasmic punctate staining, particularly Δ160p53, suggesting the formation of protein aggregates (*Figure 5C*, panels 2 and 3).

Given that p53 aggregation has been found in numerous tumor cells (*Costa et al., 2016*; *de Oliveira et al., 2015*; *Levy et al., 2011*; *Li et al., 2024*), it is important to explore the effects of Δ133p53 or Δ160p53 isoforms on the aggregation of FLp53. To investigate the aggregation property of the p53 isoforms, we performed a subcellular fractionation of the H1299 cells expressing FLp53 along with Δ133p53 or Δ160p53 at a 1:5 ratio. Notably, our fractionation approach allows us to analyze soluble cytoplasmic and nuclear fractions as well as insoluble nuclear fraction. Immunoblotting of the soluble fraction showed that an excess of Δ133p53 and Δ160p53 isoforms reduced the levels of FLp53 in both the cytoplasm and nucleus, with a notable reduction in the nucleus (*Figure 6A and C*). Conversely, insoluble nuclear FLp53 increased significantly upon co-expression with the p53 isoforms, with Δ160p53 showing a more prominent effect (*Figure 6B and D*, *Figure 6—figure supplement 1*).

To exclude the possibility that FLAG or V5 tags contribute to protein aggregation, we also conducted subcellular fractionation of H1299 cells expressing FLAG-tagged FLp53 along with untagged Δ133p53 or Δ160p53 at a 1:5 ratio. The results showed (*Figure 6—figure supplement 2*) a similar distribution of FLp53 across cytoplasmic, nuclear, and insoluble nuclear fractions as in the case of tagged Δ133p53 or Δ160p53 (*Figure 6A–D*). Notably, the aggregation of untagged Δ133p53 or Δ160p53 markedly promoted the aggregation of FLAG-tagged FLp53 (*Figure 6—figure supplement 2B and D*), demonstrating that the antibody epitope tags themselves do not contribute to protein aggregation. These results collectively demonstrate that Δ133p53 and Δ160p53 isoforms strongly induce FLp53 aggregation, and the decrease in soluble FLp53 levels could be attributed to its aggregation when co-expressed with isoforms Δ133p53 and Δ160p53.

We complemented the immunoblotting assays with immunofluorescence assays. Our results clearly show the co-localization of FLp53 with the isoforms Δ133p53 and Δ160p53. More importantly, we find co-aggregation of FLp53 with the two isoforms forming punctate spots in the cytoplasm and nucleus (*Figure 6E*, panels 2 and 3 and *Figure 6—figure supplement 3*). Furthermore, more prominent subnuclear foci of FLp53 formed in the cells co-expressing the Δ160p53 protein (*Figure 6E*, panel 3 and *Figure 6—figure supplement 3*), which were detected in the Δ133p53 protein expressing H1299 cells. Thus, these results imply that Δ133p53 and Δ160p53 isoforms efficiently induce co-aggregation of FLp53, potentially influencing its function in cancer cells.

## The Δ133p53 and Δ160p53 proteins block pro-apoptotic function of FLp53

One of the physiological read-outs of FLp53 is its ability to induce apoptotic cell death (*Aubrey et al., 2018*). To investigate the effects of p53 isoforms Δ133p53 and Δ160p53 on FLp53-induced apoptosis, we measured caspase-3 and -7 activities in H1299 cells expressing different p53 isoforms (*Figure 7*). Caspase activation is a key biochemical event in apoptosis, with the activation of effector caspases (caspase-3 and -7) ultimately leading to apoptosis (*Ghorbani et al., 2024*). The caspase-3 and -7 activities induced by FLp53 expression were approximately 2.5 times higher than that of the control vector (*Figure 7*). Co-expression of FLp53 and the isoforms Δ133p53 or Δ160p53 at a ratio of 1:5 significantly

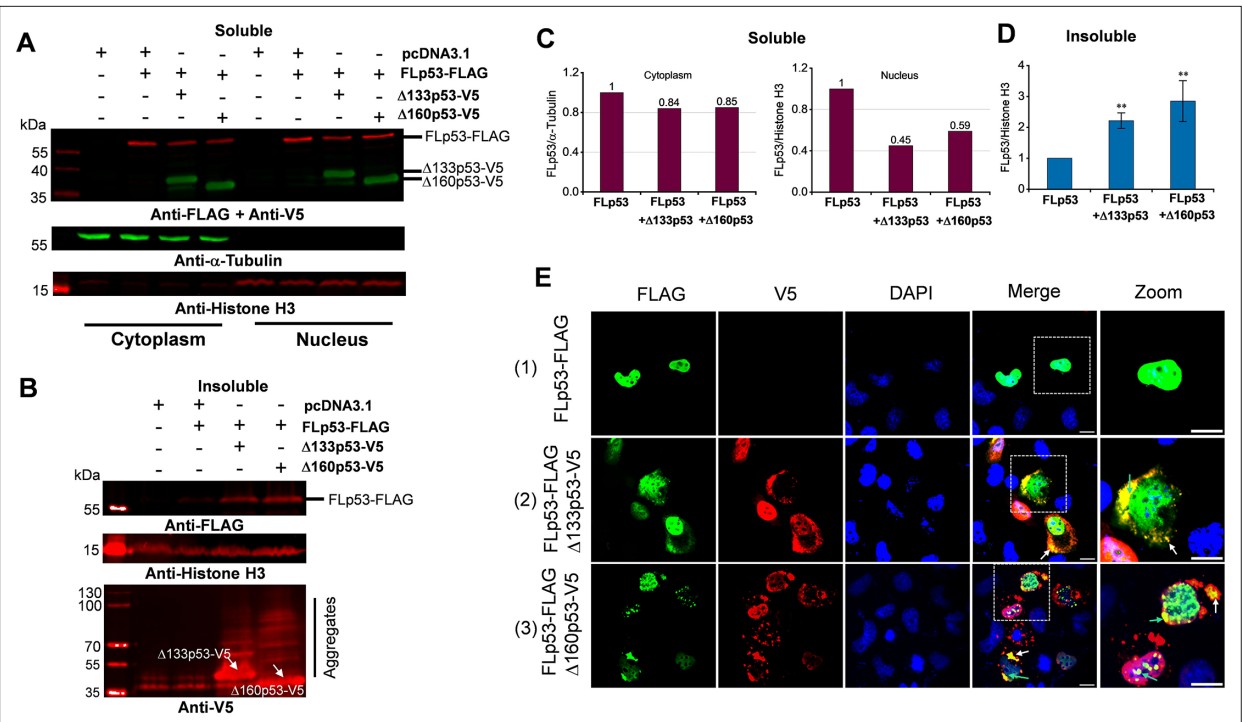

**Figure 6.** Induction of full-length p53 (FLp53) aggregation by Δ133p53 and Δ160p53 isoforms. (**A, B**) Western blot analysis of soluble cytoplasmic and nuclear subcellular fractions (**A**) and insoluble nuclear fraction (**B**). Biochemical fractionation of H1299 cells transfected with the FLAG-tagged FLp53 and V5-tagged Δ133p53 or Δ160p53 at a 1:5 ratio. The aggregated monomers of V5-tagged Δ133p53 and Δ160p53 (indicated by white arrows), as well as the higher molecular weight aggregates, are shown. This figure shows one representative experiment; n=3. (**C**) Quantitative bar graph summarizing the levels of the soluble FLp53-FLAG protein in the cytoplasmic and nuclear fractions. The FLp53-FLAG protein levels were normalized to the corresponding fractionation marker (α-Tubulin or Histone H3). Relative accumulation of the FLp53-FLAG in Δ133p53-V5 or Δ160p53-V5 expressing cells is shown after considering the FLp53-FLAG protein expressing sample as 1. (**D**) Quantitative bar graph summarizing the levels of the FLp53-FLAG protein in the insoluble nuclear fraction relative to the Histone H3. Data normalization as in panel C, data represent mean values ± standard deviation (SD). **p<0.01 (Student's t-test). (**E**) Immunofluorescence analysis of H1299 cells transfected with FLp53-FLAG alone or in combination with V5-tagged isoforms Δ133p53 and Δ160p53 expressing plasmids at a concentration ratio of 1:5. Cell nuclei were visualized with DAPI (blue). The co-aggregation of isoforms with FLp53 in the cytoplasm (white arrows) and nucleus (bright green arrows) is indicated. Scale bar, 50 µm.

The online version of this article includes the following source data and figure supplement(s) for figure 6:

**Source data 1.** Original membranes corresponding to *Figure 6A*, labeled.

**Source data 2.** Original membranes corresponding to *Figure 6B*, labeled.

**Source data 3.** Data corresponding to *Figure 6C*, showing the levels of the soluble FLp53-FLAG protein in the cytoplasmic and nuclear fractions.

**Source data 4.** Data corresponding to *Figure 6D*, showing the levels of the FLp53-FLAG protein in the insoluble nuclear fraction relative to the histone H3.

**Source data 5.** Original immunofluorescence images corresponding to *Figure 6E*.

**Source data 6.** Original membranes corresponding to *Figure 6A and B*.

**Figure supplement 1.** Western blot analysis of insoluble nuclear fraction of H1299 cells transfected with the FLAG-tagged full-length p53 (FLp53) and V5-tagged Δ133p53 or Δ160p53 at a ratio of 1:5.

**Figure supplement 1—source data 1.** Original membranes corresponding to *Figure 6—figure supplement 1*, labeled.

**Figure supplement 1—source data 2.** Original membranes corresponding to *Figure 6—figure supplement 1*.

**Figure supplement 2.** Induction of full-length p53 (FLp53) aggregation by Δ133p53 and Δ160p53 isoforms.

**Figure supplement 2—source data 1.** Original membranes corresponding to *Figure 6—figure supplement 2A*, labeled.

**Figure supplement 2—source data 2.** Original membranes corresponding to *Figure 6—figure supplement 2B*, labeled.

**Figure supplement 2—source data 3.** Data corresponding to *Figure 6—figure supplement 2C*, showing the levels of the soluble FLp53-FLAG protein in the cytoplasmic and nuclear fractions.

**Figure supplement 2—source data 4.** Data corresponding to *Figure 6—figure supplement 2D*, showing the levels of the FLp53-FLAG protein in the insoluble nuclear fraction relative to Histone H3.

*Figure 6 continued on next page*

*Figure 6 continued*

**Figure supplement 2—source data 5.** Original membranes corresponding to *Figure 6—figure supplement 2A and B*.

**Figure supplement 3.** Immunofluorescence analysis of full-length p53 (FLp53) and its isoforms Δ133p53 and Δ160p53 localization in H1299 cells.

**Figure supplement 3—source data 1.** Original immunofluorescence images corresponding to *Figure 6—figure supplement 3*.

diminished the apoptotic activity of FLp53 (*Figure 7*). This result aligns well with our reporter gene assay, which demonstrated that elevated expression of Δ133p53 and Δ160p53 impaired the expression of apoptosis-inducing genes BAX and PUMA (*Figure 4G and H*). Moreover, a reduction in the apoptotic activity of FLp53 was observed irrespective of whether Δ133p53 or Δ160p53 protein was expressed with or without a FLAG tag (*Figure 7*). This result, therefore, also suggests that the FLAG tag does not affect the apoptotic activity or other physiological functions of FLp53 and its isoforms. Overall, the overexpression of p53 isoforms Δ133p53 and Δ160p53 significantly attenuates FLp53-induced apoptosis, independent of the protein tagging with the FLAG antibody epitope.

## Discussion

The inactivation of p53 is one of the most common events in cancer cells (*Kussie et al., 1996*). Although functionally compromised p53 mutations are prevalent in more than half of human cancers (*Bykov et al., 2018*; *Hollstein et al., 1991*), specific p53 isoforms have recently been associated with tumorigenesis in various cancer types (*Vieler and Sanyal, 2018*; *Zhao and Sanyal, 2022*). The aberrant expression of Δ133p53 has been found in colorectal cancer (*Campbell et al., 2018*; *Fujita et al., 2009*) and lung cancer (*Fragou et al., 2017*). In comparison, the oncogenic role of the Δ160p53 isoform in human cancers is less explored (*Zhao and Sanyal, 2022*), although links have been established between the expression of the Δ160p53 isoform and melanoma cells (*Tadijan et al., 2021*). The mechanisms of the oncogenic potential of p53 isoforms have attracted attention from researchers (*Arsic et al., 2021*; *Gong and Chen, 2016a*; *Lei et al., 2019*; *Melo Dos Santos et al., 2019*; *Mondal et al., 2013*; *Vieler and Sanyal, 2018*). A theory of the dominant-negative effect has been introduced, but its exact details have remained unknown.

The ChIP and luciferase activity assays show that the isoforms Δ133p53 and Δ160p53 are defective in their ability to bind DNA and activate transcription (*Figures 3B and 4A–D*). A possible reason for the diminished DNA-binding function of these isoforms is that their core domains are defective (*Figure 1B*). The Δ133p53 and Δ160p53 isoforms lack approximately 20% and 33% of the DBD, which hinders their ability to bind to p53REs. We find that co-expression of Δ133p53 and Δ160p53 with FLp53 exerts an inhibitory effect on FLp53's DNA-binding ability (*Figure 3C*). However, the degree to which Δ133p53 and Δ160p53 impair FLp53 function is influenced by both the expression levels of these isoforms and the specific promoters of the target genes (*Figure 4*). For the p21 and MDM2 promoters, equal expression of FLp53 and the two isoforms didn't show much effect. However, higher expression levels of the isoforms gradually inhibited FLp53's activity to induce transcription from those promoters (*Figure 4A and B*). Notably, the PUMA promoter behaved differently, whereby 1:1 co-expression of FLp53 with Δ133p53 or Δ160p53 significantly

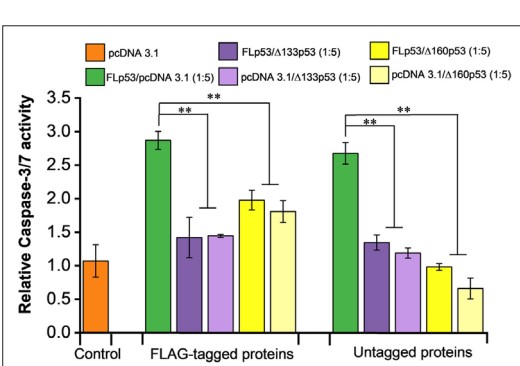

**Figure 7.** Overexpression of Δ133p53 and Δ160p53 blocks pro-apoptotic activity of full-length p53 (FLp53). Caspase-3 and -7 activities were detected in H1299 cells co-transfected with plasmids expressing FLp53 and either Δ133p53 or Δ160p53 at a 1:5 ratio. Experiments were performed with the plasmids expressing FLAG-tagged and untagged FLp53, Δ133p53, and Δ160p53. The relative activities of caspase-3 and -7 were normalized to those of the control vector (transfected only with the pcDNA3.1 plasmid). Data represent mean values ± standard deviation (SD). **p<0.01 (Student's t-test).

The online version of this article includes the following source data for figure 7:

**Source data 1.** Data corresponding to *Figure 7*, showing caspase-3 and -7 activities of H1299 cells co-transfected with plasmids expressing FLp53 and either Δ133p53 or Δ160p53 at a 1:5 ratio.

increased its activity (*Figure 4D*). We speculate that this isoform expression level may activate p21 induction pathways independent of FLp53 (*Karimian et al., 2016*), potentially facilitating the interaction between FLp53 and p21, which is required for optimal activation of PUMA promoter activity (*Kim et al., 2022*). These results underscore a complex regulatory network where FLp53 and the isoform expression levels, target gene specificity, and transcriptional activities modulate p53-mediated cellular responses.

Our co-expression data clearly indicate that the isoforms Δ133p53 and Δ160p53 exert a dominant-negative effect on FLp53 function (*Figures 3, 4, and 7*). Since both of these p53 isoforms are defective in DNA binding and they possess an intact C-terminal OD, it is not unexpected that they form different species of homo- and hetero-tetramers with FLp53. In accordance with that, the immunoprecipitation assays using FLAG-tagged FLp53 and V5-tagged/untagged Δ133p53 or Δ160p53 demonstrate a potential direct interaction between these isoforms and FLp53 (*Figure 2A*, lanes 7 and 8 and *Figure 2—figure supplement 1*, lanes 3 and 4). However, in what ratio the p53 isoforms impose inhibition to FLp53 function is worth investigating. We employed a theoretical model of p53 hetero-tetramer activity in transcriptional activation from p21, MDM2, BAX, and PUMA promoters by varying relative expression levels of the isoforms (*Chan et al., 2004*; *Figure 4—figure supplement 1*). We assumed that the transfection level was a good proxy for the expression level. Interestingly, the experimental inhibition curves with Δ133p53 or Δ160p53 derived from the luciferase reporter assay were all above the theoretical inhibition curves for two isoform molecules per tetramer (*Figure 4—figure supplement 1B–E*), suggesting that at least three isoform molecules per tetramer are required to abolish the transcriptional activity of FLp53. This is similar to an earlier report with Δ40p53 isoform, which was shown to exert a dominant-negative effect at a 3:1 ratio with FLp53 (*Hafsi et al., 2013*).

Interestingly, similar to Δ40p53 (*Melo Dos Santos et al., 2019*), the Δ133p53 and Δ160p53 proteins can aggregate in the cytoplasm (*Figures 5C and 6E*). Due to cytoplasmic aggregation and a stronger tendency for cytoplasmic localization (*Figure 5*), only a limited amount of these isoforms can successfully be imported into the nucleus. Additionally, the isoforms tend to aggregate within the nucleus (*Figure 6B and E*). Consequently, the ratio of these isoforms to FLp53 diminishes in the nucleus in contrast to their expression levels. This decrease reduces the likelihood of forming hetero-tetramers, in which the isoforms will be in higher proportion than FLp53. Thus, it seems unlikely that hetero-tetramerization of Δ133p53 and Δ160p53 isoforms with FLp53 would be the main reason for their dominant-negative effect.

Next, we investigated whether aggregation of the Δ133p53 and Δ160p53 isoforms contributes to their functional loss and dominant-negative effect on FLp53. Our immunofluorescence and immunoprecipitation analyses of H1299 cells expressing these isoforms show a higher aggregation propensity of the two isoforms than FLp53 (*Figure 5C*, *Figure 1—figure supplement 2*). The aggregation likely results from the exposure of hydrophobic sequences in the isoforms (*Figure 1C*). In addition, the absence of N-terminal TADs in Δ133p53 and Δ160p53 isoforms may prevent their MDM2-mediated ubiquitin-proteasome degradation (*Chen et al., 1993*; *Kussie et al., 1996*). This may, in turn, lead to higher accumulation of Δ133p53 and Δ160p53 isoforms in the cell, which, combined with an unstable core domain, promotes their aggregation. Moreover, the observed cytoplasmic isoform aggregates may reflect autophagy-related degradation, as suggested by the co-localization of Δ133p53 with autophagy substrate p62/SQSTM1 and autophagosome component LC3B (*Horikawa et al., 2014*). Overexpression of these aggregation-prone proteins could induce endoplasmic reticulum stress and activate autophagy (*Lee et al., 2012*). Interestingly, we also observed nuclear aggregation of these isoforms (*Figure 6B*, *Figure 6—figure supplement 2*), suggesting that distinct mechanisms, such as intrinsic properties of the isoforms, may govern their localization and behavior within the nucleus. This dual localization underscores the complexity of Δ133p53 and Δ160p53 behavior in cellular systems.

More interestingly, our immunofluorescence and immunoprecipitation assays demonstrate large co-aggregation of FLp53 with Δ133p53 and Δ160p53 isoforms in the cytoplasm as well as in the nucleus when expressed together (*Figure 6E*, *Figure 6—figure supplement 3*). The predominant cytoplasmic aggregation would sequester FLp53 in the cytoplasm. Moreover, it may destabilize FLp53's conformation, potentially masking the NLS and preventing its nuclear import (*Marine, 2010*). Additionally, it may disrupt the interaction of FLp53 with MDM2, thereby inhibiting its MDM2-induced degradation and enhancing its accumulation (*Moll and Petrenko, 2003*), which would drive it more to the aggregation pathway. All these processes involving FLp53 aggregation would result in a reduction

of the functional FLp53 in the cell and, more importantly, in the nucleus. Cytoplasmic aggregation would also limit the isoforms from nuclear import, reducing the possibility of hetero-tetramerization. Therefore, we propose that the co-aggregation of Δ133p53 and Δ160p53 with FLp53 is the primary factor for their dominant-negative effect. This mechanism is consistent with the model where structural mutations, such as R175H and R282W in FLp53, induce co-aggregation and exert a dominant-negative effect (*Xu et al., 2011*).

In our study, we primarily utilized an overexpression strategy involving FLAG/V5-tagged proteins to investigate the effects of p53 isoforms Δ133p53 and Δ160p53 on the function of FLp53. To address concerns regarding potential overexpression artifacts, we performed the co-immunoprecipitation (*Figure 6—figure supplement 2*) and caspase-3 and -7 activity (*Figure 7*) experiments with untagged Δ133p53 and Δ160p53. In both experimental systems, the untagged proteins behaved very similarly to the FLAG/V5 antibody epitope-containing proteins (*Figures 6 and 7*, *Figure 6—figure supplement*

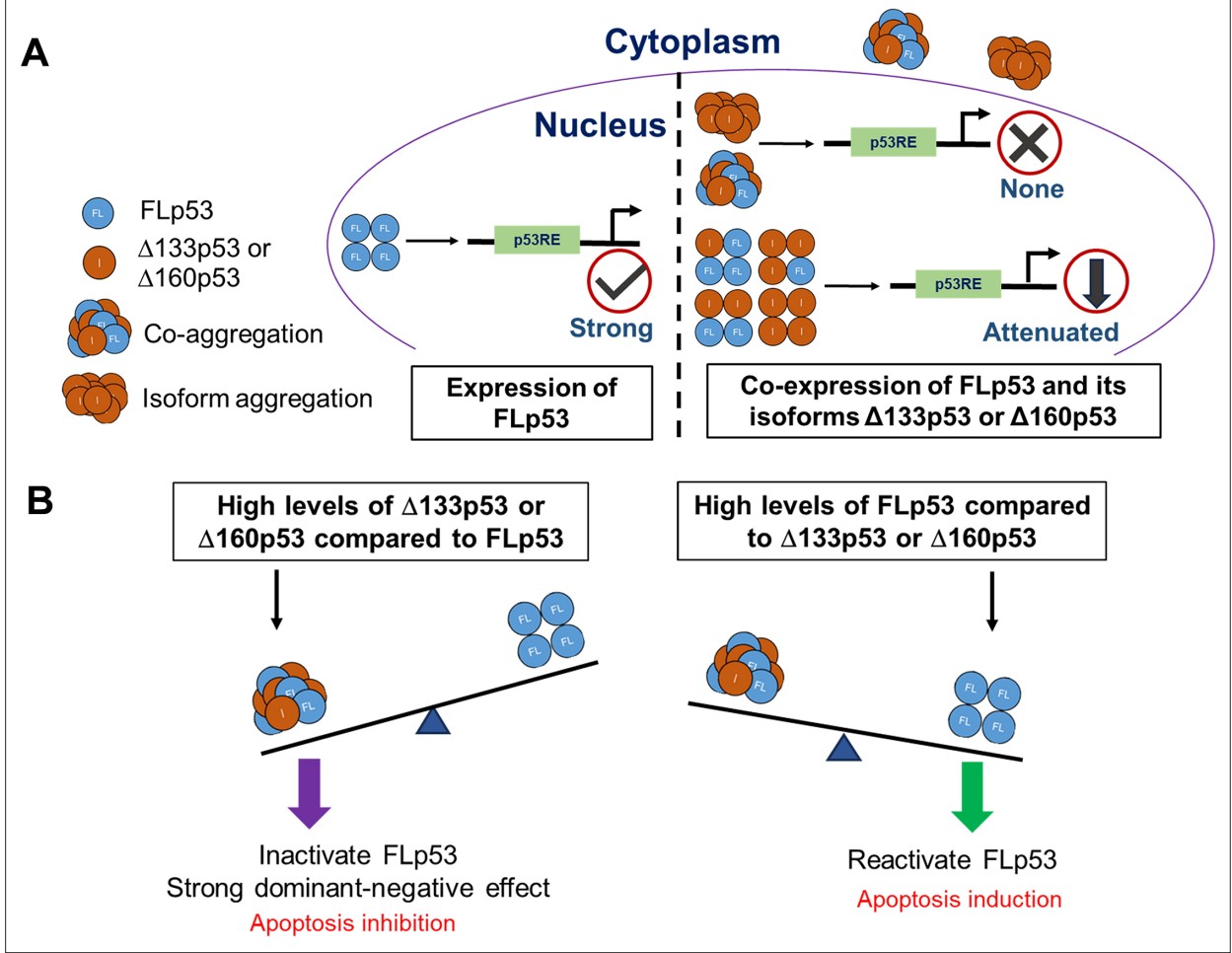

**Figure 8.** Schematic representation of the underlying mechanisms by which Δ133p53 and Δ160p53 exert a dominant-negative effect on full-length p53 (FLp53). (**A**) When expressed alone, FLp53 demonstrates strong transcriptional activity, but when expressed together with the Δ133p53 or Δ160p53 isoforms, FLp53 activity gets impaired due to the dominant-negative effect exerted by the isoforms. The Δ133p53 or Δ160p53 isoforms either attenuate FLp53's transcription activity by hetero-tetramerization at a ratio higher than 1:1 or abolish it completely by aggregation. Since Δ133p53 and Δ160p53 exhibit much higher propensity for aggregation than FLp53, they sequester FLp53 in the cytoplasmic aggregates, leading to a reduction in the amount of functional FLp53 in the nucleus. Aggregation also limits the availability of these isoforms in the nucleus. Thus, co-aggregation is the primary reason for the dominant-negative effect of the Δ133p53 or Δ160p53 isoforms on FLp53. It is important to note that the numbers of tetramers and aggregates depicted do not represent the actual quantities present in the cytoplasm and nucleus. (**B**) The balance between the expression level of the FLp53 and its isoforms Δ133p53 and Δ160p53 regulates FLp53 activity. Increased levels of Δ133p53 and Δ160p53 relative to FLp53 lead to a stronger dominant-negative effect due to co-aggregation and hetero-tetramerization, resulting in the inhibition of apoptosis. Increased levels of FLp53 compared to the isoforms Δ133p53 and Δ160p53 in cancer cells restore the function of FLp53, leading to the induction of apoptosis.

*2*). Hence, the C-terminal tagging of FLp53 or its isoforms does not alter the biochemical and physiological functions of these proteins.

We propose a model to explain how the p53 isoforms Δ133p53 and Δ160p53 inhibit FLp53 activity and exert a dominant-negative effect on it (*Figure 8A*). Two mechanisms are involved: a subsidiary mechanism in which Δ133p53 and Δ160p53 isoforms form hetero-tetramers with FLp53, impairing its DNA-binding and transcriptional activities, and a predominant mechanism in which these isoforms promote co-aggregation with FLp53 and abolish its transcriptional activities by disrupting its structure, regulation, and degradation pathways. Since elevated levels of Δ133p53 and Δ160p53 relative to FLp53 attenuate its transcriptional activity (*Figure 8B*, left), and overexpression of FLp53 relative to the isoforms allows regaining it (*Figure 8B*, right), the tumor-suppressive efficacy of p53 must depend on the balanced expression of FLp53 and its isoforms, highlighting the importance of their relative abundance in cancer cells. Based on our results and recent related publications (*Arsic et al., 2021*; *Melo Dos Santos et al., 2019*), we propose that co-aggregation, rather than hetero-tetramerization, is the primary mechanism for the dominant-negative effect exerted by the p53 isoforms with large N-terminal truncations.

## Materials and methods
### Cell lines, plasmids, and plasmid transfections

Human non-small cell lung carcinoma cells, NCI-H1299 cells, were from ATCC (RRID:CVCL_0060) and were also authenticated using STR profiling (Eurofins Genomics). Cells were routinely tested for mycoplasma contamination (Mycoplasmacheck, Eurofins Genomics). H1299 cells were cultured in Dulbecco's modified Eagle's medium supplemented with 10% fetal bovine serum (Thermo Fisher Scientific Cat# A5256701) and penicillin-streptomycin solution (PEST; Thermo Fisher Scientific Cat# 15140122) at 37°C in a 5% $CO_2$ incubator. The plasmid pcDNA3.1-FLp53-FLAG, encoding human FLp53 (amino acids 1–393), was synthesized by GenScript (Rijswijk, Netherlands). A flexible linker $(GGGGG)_1$ was introduced between the FLp53 coding sequence and the C-terminal FLAG epitope tag. Subsequently, this fusion construct was inserted into the pcDNA3.1 vector (RRID:Addgene_20140) at the NheI and EcoRI restriction enzyme sites. The plasmids encoding N-terminally truncated p53 proteins (Δ133p53 and Δ160p53) were created by deleting the N-terminal 132 and 159 amino acids from the FLp53 sequence, respectively. To create plasmids expressing C-terminal V5-tagged proteins (FLp53, Δ133p53, and Δ160p53), the respective DNA sequences were amplified by PCR from the pcDNA3.1-FLp53-FLAG template and inserted into the EcoRI and XhoI sites of the pcDNA3.1-V5. The plasmids encoding untagged p53 proteins were constructed by inserting the respective DNA sequences into the NheI and EcoRI restriction sites of the pcDNA3.1 empty vector. Luciferase reporter plasmids, WWP/p21-Luc (RRID:Addgene_16451) (*el-Deiry et al., 1993*), pGL3-MDM2-Luc (RRID:Addgene_32369) (*Chang et al., 2004*), and PUMA Frag1-Luc (RRID:Addgene_16591) (*Yu et al., 2001*) were obtained from Addgene. For the construction of the BAX-Luc reporter plasmid, the p53 binding site in the BAX promoter was PCR-amplified from HepG2 genomic DNA kindly provided by Dr. Gang Pan (Uppsala University) and cloned into the pGL3-Basic plasmid (RRID:Addgene_212936) using XhoI and HindIII sites. The PCR primers used for cloning are listed in *Supplementary file 1*. All plasmid transfections were performed with jetPRIME (Polyplus) transfection reagent according to the manufacturer's protocol.

### Immunoprecipitation and western blotting

H1299 cells were seeded in a six-well plate and transfected with 2 µg of various plasmids. The total amount of plasmids (2 µg/well) was kept constant in all samples by adding empty plasmid pcDNA3.1. To investigate p53 oligomerization, cells were co-transfected with equal amounts (1 µg each) of FLAG- and V5-tagged plasmids. After 24 hr post-transfection (hpt), cells were harvested and lysed in RIPA buffer (50 mM Tris-HCl pH 7.4, 150 mM NaCl, 1% Triton X-100, 1% sodium deoxycholate, 0.1% SDS, and 1 mM EDTA) supplemented with the Halt Protease Inhibitor Cocktail (Thermo Fisher Scientific Cat# 78440) as previously described (*Inturi et al., 2013*). The whole-cell lysates were incubated for 30 min on ice and centrifuged at 13,000×*g* for 15 min at 4°C. A fraction of the cleared cell lysates was reserved as the input sample; the remaining lysates were used for immunoprecipitation with anti-FLAG M2 magnetic beads (Sigma-Aldrich Cat# M8823, RRID:AB_2637089) overnight at 4°C. The

beads were washed 4× with 1 ml of RIPA buffer. Proteins were then eluted from the beads using 2× Laemmli Sample buffer, separated on a 10% SDS-PAGE, transferred onto nitrocellulose membranes (Bio-Rad Cat# 1620112), and blocked with 5% nonfat dry milk in TBS containing 0.1% Tween-20 (TBST) for 1 hr at room temperature. The blocked membranes were incubated overnight at 4°C with the anti-FLAG (Sigma-Aldrich Cat# F2555, RRID:AB_796202) and anti-V5 (Sigma-Aldrich Cat# V8012, RRID:AB_261888) primary antibodies in a blocking buffer overnight at 4°C. Proteins were detected with the IRDye 680RD Goat anti-Rabbit IgG and IRDye 800CW Donkey anti-Mouse IgG secondary antibodies (LI-COR Biosciences). The western blots were visualized using an Odyssey CLX imaging system (LI-COR Biosciences).

## ChIP assays

H1299 cells were seeded in 10 cm cell culture dishes and transfected with 5 µg of either pcDNA3.1-FLp53-FLAG, pcDNA3.1-Δ133p53-FLAG, pcDNA-Δ160p53-FLAG, or an empty vector pcDNA3.1 for 24 hr. To investigate the impacts of isoforms Δ133p53 and Δ160p53 on the DNA-binding activity of FLp53, H1299 cells were co-transfected with 2.5 µg of pcDNA3.1-FLp53-FLAG along with either 2.5 µg of pcDNA3.1-Δ133p53-V5, pcDNA3.1-Δ160p53-V5, or an equal amount of empty vector pcDNA3.1. The ChIP assays were performed as described previously (*Punga and Bühler, 2010*), with minor modification. Briefly, cells were fixed with 1% formaldehyde in PBS for 10 min at room temperature, followed by lysis in 250 µl of lysis buffer (50 mM Tris-HCl pH 8.0, 10 mM EDTA, 1% SDS) supplemented with Halt Protease Inhibitor Cocktail. Lysates were then sonicated for 15 cycles (30 s ON/30 s OFF) using the Bioruptor Pico (Diagenode Cat# B01080010). The resulting soluble cell lysates were diluted with dilution buffer (16.7 mM Tris-HCl pH 8.0, 167 mM NaCl, 1.2 mM EDTA, 0.01% SDS, 1.1% Triton X-100) and immunoprecipitated overnight at 4°C with the anti-FLAG M2 magnetic beads. After a series of washes, protein-DNA complexes were eluted, and the cross-links were reversed through incubation with RNase A (Thermo Fischer Scientific Cat# EN0531) at 37°C for 30 min, followed by proteinase K (Invitrogen Cat# 25530049) digestion at 65°C overnight. The enrichment of specific DNA fragments was evaluated with qPCR analysis using HOT FIREPol EvaGreen qPCR Supermix (Solis Biodyne Cat# 08-36-00001). All qPCRs were done in triplicate, employing the QuantStudio 6 Flex Real-Time PCR System (Thermo Fisher Scientific Cat# 4485691). The relative fold enrichments were calculated after normalizing to the input values and considering the pcDNA3.1 transfected samples as 1. Primer sequences used for qPCR analysis are listed in *Supplementary file 1*.

## Luciferase assay

H1299 cells were seeded in 24-well plates and transfected in triplicate with 25 ng/well of luciferase reporter plasmids and various p53-expressing plasmids. Co-transfection ratios for FLp53 and its isoforms Δ133p53 or Δ160p53 were set at 1:1 (12.5 ng/well of FLp53 to 12.5 ng/well of isoform), 1:5 (12.5 ng/well of FLp53 to 62.5 ng/well of isoform), and 1:10 (12.5 ng/well of FLp53 to 125 ng/well of isoform). The total plasmid amount (300 ng/well) was kept consistent across the various groups by supplementing with the empty pcDNA3.1 plasmid. Cells were harvested 24 hpt and lysed in a Passive Lysis Buffer (Promega Cat# E194A). The cell lysate was centrifuged at 13,000×*g* for 5 min at 4°C, and 20 µl of the lysate was analyzed in triplicates using a firefly luciferase substrate on Infinite 200 PRO plate reader (TECAN) (*Schubert et al., 2024*). The luciferase activity of the FLp53 and its isoforms-transfected samples was normalized against that of samples transfected with the control vector pcDNA3.1 to obtain relative reporter activity. Results represent the mean values and standard deviations of three independent experimental replicates. The data was analyzed using Origin 2018 software (Origin Lab Corporation).

## Subcellular fractionation

The subcellular fractionation assay was carried out using NE-PER Nuclear and Cytoplasmic Extraction Reagents (Thermo Fisher Scientific Cat# 78833), with minor modifications. In brief, H1299 cells were seeded in 10 cm cell culture dishes and transfected with 0.8 µg of the following plasmids: pcDNA3.1-FLp53-FLAG, pcDNA3.1-Δ133p53-FLAG, pcDNA-Δ160p53-FLAG, and the control vector pcDNA3.1. To investigate the effects of overexpressing p53 isoforms on the subcellular localization of FLp53, co-transfection was performed using FLAG-tagged FLp53 and V5-tagged Δ133p53 or Δ160p53 at a 1:5 ratio (0.8–4.2 µg). Cells were transfected with 5 µg of the empty vector pcDNA3.1 as a control.

About 24 hpt, cells were collected by trypsinization, resuspended in ice-cold CER I buffer containing Halt Protease Inhibitor Cocktail, and incubated for 10 min on ice. Subsequently, ice-cold CER II buffer was added according to the manufacturer's instructions. The cytoplasmic extracts were isolated by centrifugation at 14,000×*g* for 5 min at 4°C. The nuclear pellet was then resolved in ice-cold RIPA buffer supplemented with Halt Protease Inhibitor Cocktail, subjected to vortex for 15 s every 10 min for a total of 40 min, and centrifuged at 14,000×*g* for 10 min to obtain the soluble nuclear fraction. The insoluble nuclear pellet (i.e. insoluble nuclear fraction) was resuspended in RIPA buffer, an appropriate volume of 2× Laemmli Sample buffer was added, and the samples were heated at 95°C until the pellet fully dissolved. For western blot experiments, equal volumes of each subcellular fraction were separated by SDS-PAGE, and the nitrocellulose membranes were probed with the respective antibodies. Histone H3 (nuclear marker) and α-Tubulin (cytoplasmic marker) antibodies were used to verify the purity of the nuclear and cytoplasmic fractions. All primary antibodies: anti-FLAG (Sigma-Aldrich Cat# F2555, RRID:AB_796202), anti-V5 (Sigma-Aldrich Cat# V8012, RRID:AB_261888 and Sigma-Aldrich Cat# V8137, RRID:AB_261889), anti-Histone H3 (Abcam Cat# ab1791, RRID:AB_302613), and anti-α-Tubulin (Sigma-Aldrich Cat# T6199, RRID:AB_477583) were incubated on membranes overnight at 4°C. The western blots were visualized using an Odyssey CLX imaging system, and the proteins were quantified using Image Studio Version 5.5 (LI-COR).

## Immunofluorescence analysis

H1299 cells were seeded on glass coverslips in 24-well plates and transfected with 250 ng/well of either pcDNA3.1-FLp53-FLAG, pcDNA3.1-Δ133p53-FLAG, pcDNA-Δ160p53-FLAG, or the control vector pcDNA3.1. To investigate whether an excess of p53 isoforms could influence the subcellular localization of FLp53, H1299 cells were co-transfected with 250 ng/well of pcDNA3.1-FLp53-FLAG along with 1000 ng/well of pcDNA3.1-Δ133p53-V5, pcDNA3.1-Δ160p53-V5, or pcDNA3.1. About 24 hpt, cells were fixed with 4% paraformaldehyde in PBS, followed by permeabilization with 0.2% Triton X-100 in PBS for 15 min at room temperature as described previously (*Kases et al., 2023*). Fixed cells on coverslips were incubated overnight at 4°C with primary antibodies anti-FLAG (diluted 1:1000, Sigma-Aldrich Cat# F1804, RRID:AB_262044) and anti-V5 (diluted 1:1000, Sigma-Aldrich Cat# V8137, RRID:AB_261889). Subsequently, coverslips were incubated with the Alexa Fluor 488 goat anti-mouse IgG (diluted 1:2000, Thermo Fisher Scientific Cat# A-11001, RRID:AB_2534069) and/or Alexa Fluor 594 chicken anti-rabbit IgG (H+L) (diluted 1:2000, Thermo Fisher Scientific Cat# A-21442, RRID:AB_2535860) secondary antibodies for 30 min at room temperature. The coverslips were mounted with ProLong Diamond Antifade Mountant (Invitrogen Cat# P36961), and nuclei were counterstained using DAPI. Subcellular localization was visualized using a Nikon Eclipse 90i microscope (Nikon, Japan). Images were processed using ImageJ software (*Schindelin et al., 2012*) and subsequently arranged by Inkscape.

## Caspase-3/7 activity assay

Caspase-3 and -7 activities were measured with a Caspase-Glo 3/7 assay kit (Promega Cat# G8090) according to the manufacturer's protocol. Briefly, H1299 cells were seeded (5000 cells/well) into a white solid bottom 96-well plate and transfected in triplicate with 180 ng/well of plasmids expressing FLAG-tagged or untagged FLp53, Δ133p53, Δ160p53, or the empty vector control pcDNA3.1. To evaluate the effects of overexpression of isoforms on the apoptotic activity of FLp53, H1299 cells were co-transfected with plasmids encoding FLp53 (30 ng/well) with the control plasmid pcDNA3.1 (150 ng/well), or plasmids expressing isoforms Δ133p53 and Δ160p53 (150 ng/well), at a 1:5 ratio. After 28 hpt, caspase-3/7 substrate was added to each well and incubated for 90 min at room temperature. Luminescence was recorded using an Infinite 200 PRO plate reader (TECAN). The background luminescence associated with the cell culture and assay reagent (blank reaction) was subtracted from the experimental values.

## Acknowledgements

We thank Dr. Zamaneh Hajikhezri for her help with the immunofluorescence experiments and Dr. Gang Pan for reagents. This research was funded by grants from the Swedish Research Council (2023-05237, 2018-05946, and 2018-05498), from Wenner-Gren Foundation (UPD2023-0185) and from the Knut and Alice Wallenberg Foundation (KAW 2017.0055) to SS.

## Additional information

### Funding

| Funder | Grant reference number | Author |
|---|---|---|
| Vetenskapsrådet | 2023-05237 | Suparna Sanyal |
| Vetenskapsrådet | 2018-05946 | Suparna Sanyal |
| Vetenskapsrådet | 2018-05498 | Suparna Sanyal |
| Wenner-Gren Stiftelserna | UPD2023-0185 | Suparna Sanyal |
| Knut och Alice Wallenbergs Stiftelse | KAW 2017.0055 | Suparna Sanyal |

The funders had no role in study design, data collection and interpretation, or the decision to submit the work for publication.

### Author contributions

Liuqun Zhao, Data curation, Formal analysis, Investigation, Writing – original draft, Writing – review and editing; Tanel Punga, Data curation, Formal analysis, Supervision, Visualization, Methodology, Writing – review and editing; Suparna Sanyal, Conceptualization, Formal analysis, Supervision, Funding acquisition, Investigation, Project administration, Writing – review and editing, Resources

### Author ORCIDs

Liuqun Zhao (iD) https://orcid.org/0000-0002-0948-0923
Tanel Punga (iD) https://orcid.org/0000-0002-0561-367X
Suparna Sanyal (iD) https://orcid.org/0000-0002-7124-792X

Reviewer #1 (Public review): https://doi.org/10.7554/eLife.106469.2.sa1
Reviewer #2 (Public review): https://doi.org/10.7554/eLife.106469.2.sa2
Author response https://doi.org/10.7554/eLife.106469.2.sa3

# Additional files

### Supplementary files

Supplementary file 1. Sequences of oligonucleotides used in the work.

MDAR checklist

### Data availability

All data generated or analysed during this study are included in the manuscript and supporting files; source data files have been provided for Figures 2–7, Figure 1—figure supplement 2, Figure 2—figure supplement 1, Figure 5—figure supplement 1, and Figure 6—figure supplement 1–3. All source data files are available in Zenodo Data Platform (Dataset DOI: https://doi.org/10.5281/zenodo.15688023).

The following dataset was generated:

| Author(s) | Year | Dataset title | Dataset URL | Database and Identifier |
|---|---|---|---|---|
| Zhao L, Pung T, Sanyal S | 2025 | Δ133p53α and Δ160p53α isoforms of the tumor suppressor protein p53 exert dominant-negative effect primarily by co-aggregation | https://doi.org/10.5281/zenodo.15688023 | Zenodo, 10.5281/zenodo.15688023 |

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
