## [Editor Report · eLife Assessment]

This **important** study investigates the molecular mechanisms by which the p53 isoforms Δ133p53α and Δ160p53α exert dominant-negative effects on full-length p53 (FLp53). Through a combination of chromatin immunoprecipitation, transcriptional reporter assays, subcellular localization analyses, and protein aggregation experiments, the authors provide **solid** evidence that these N-terminally truncated isoforms promote co-aggregation with FLp53, disrupting its transcriptional activity and cellular distribution. The revised manuscript successfully addresses prior reviewer concerns, and the findings are well supported by the experimental data.

[Editors' note: this paper was reviewed by Review Commons.]

---

## [Referee Report · Reviewer #1 (Public review)]

Summary:

The authors have provided a mechanism by which how presence of truncated P53 can inactivate function of full length P53 protein. The authors proposed this happens by sequestration of full length P53 by truncated P53. In the study, the performed experiments are well described.

Significance:

The work in significant, since it points out more mechanistic insight how wild type full length P53 could be inactivated in the presence of truncated isoforms, this might offer new opportunity to recover P53 function as treatment strategies against cancer.

Comments on latest version:

The authors have made significant effort to address my concerns using the system available to them. I find the justifications provided in the rebuttal letter and the revised figures satisfactory. My initial concerns regarding the overexpression system have been largely addressed. However, the experimental system used by the authors lacks the means to measure the effect on endogenous p53, which remains a limitation.

---

## [Referee Report · Reviewer #2 (Public review)]

Summary:

The revised manuscript by Zhao and colleagues presents a novel and compelling investigation into the p53 isoforms, Δ133p53 and Δ160p53, which are implicated in aggressive cancer phenotypes. The primary goal of this study was to elucidate how these isoforms exert a dominant-negative impact on the activity of full-length p53 (FLp53). The authors demonstrate that the Δ133p53 and Δ160p53 isoforms display impaired binding to p53-regulated promoters. Their findings suggest that the dominant-negative effects observed are primarily due to the co-aggregation of FLp53 with Δ133p53 and Δ160p53.

Overall, the study is innovative, thoroughly executed, and supported by robust data analysis. The authors have effectively addressed the reviewers' criticisms and incorporated their suggestions in this revised manuscript.

Significance:

The manuscript by Zhao and colleagues presents a novel and compelling study on the p53 isoforms, Δ133p53 and Δ160p53, which are associated with aggressive cancer types. The main objective of the study was to understand how these isoforms exert a dominant negative effect on full-length p53 (FLp53). The authors discovered that the Δ133p53 and Δ160p53 proteins exhibit impaired binding to p53-regulated promoters. The data suggest that the predominant mechanism driving the dominant-negative effect is the co-aggregation of FLp53 with Δ133p53 and Δ160p53.

---

## [Author Response]

**Reviewer #1 (Evidence, reproducibility and clarity):**
Authors has provided a mechanism by which how presence of truncated P53 can inactivate function of full length P53 protein. Authors proposed this happens by sequestration of full length P53 by truncated P53.In the study, performed experiments are well described.My area of expertise is molecular biology/gene expression, and I have tried to provide suggestions on my area of expertise. The study has been done mainly with overexpression system and I have included few comments which I can think can be helpful to understand effect of truncated P53 on endogenous wild type full length protein. Performing experiments on these lines will add value to the observation according to this reviewer.Major comments:(1) What happens to endogenous wild type full length P53 in the context of mutant/truncated isoforms, that is not clear. Using a P53 antibody which can detect endogenous wild type P53, can authors check if endogenous full length P53 protein is also aggregated as well? It is hard to differentiate if aggregation of full length P53 happens only in overexpression scenario, where lot more both of such proteins are expressed. In normal physiological condition P53 expression is usually low, tightly controlled and its expression get induced in altered cellular condition such as during DNA damage. So, it is important to understand the physiological relevance of such aggregation, which could be possible if authors could investigate effect on endogenous full length P53 following overexpression of mutant isoforms.

Thank you very much for your insightful comments.

(1) To address “what happens to endogenous wild-type full-length P53 in the context of mutant/truncated isoforms," we employed a human A549 cell line expressing endogenous wild-type p53 under DNA damage conditions such as an etoposide treatment(1). We choose the A549 cell line since similar to H1299, it is a lung cancer cell line (https://www.atcc.org/). For comparison, we also transfected the cells with 2 μg of V5-tagged plasmids encoding FLp53 and its isoforms Δ133p53 and Δ160p53. As shown in Author response image 1A, lanes 1 and 2, endogenous p53 expression, remained undetectable in A549 cells despite etoposide treatment, which limits our ability to assess the effects of the isoforms on the endogenous wild-type FLp53. We could, however, detect the V5-tagged FLp53 expressed from the plasmid using anti-V5 (rabbit) as well as with antiDO-1 (mouse) antibody (Author response image 1). The latter detects both endogenous wildtype p53 and the V5-tagged FLp53 since the antibody epitope is within the Nterminus (aa 20-25). This result supports the reviewer’s comment regarding the low level of expression of endogenous p53 that is insufficient for detection in our experiments.

In summary, in line with the reviewer’s comment that ‘under normal physiological conditions p53 expression is usually low,’ we could not detect p53 with an anti-DO-1 antibody. Thus, we proceeded with V5/FLAG-tagged p53 for detection of the effects of the isoforms on p53 stability and function. We also found that protein expression in H1299 cells was more easily detectable than in A549 cells (Compare Author response image 1A and B). Thus, we decided to continue with the H1299 cells (p53-null), which would serve as a more suitable model system for this study.

(2) We agree with the reviewer that ‘It is hard to differentiate if aggregation of full-length p53 happens only in overexpression scenario’. However, it is not impossible to imagine that such aggregation of FLp53 happens under conditions when p53 and its isoforms are over-expressed in the cell. Although the exact physiological context is not known and beyond the scope of the current work, our results indicate that at higher expression, p53 isoforms drive aggregation of FLp53. Given the challenges of detecting endogenous FLp53, we had to rely on the results obtained with plasmid mediated expression of p53 and its isoforms in p53-null cells.

**Author response image 1. sa3fig1:** Comparative analysis of protein expression in A549 and H1299 cells. (A) A549 cells (p53 wild-type) were treated with etoposide to induce endogenous wild-type p53 expression. To assess the effects of FLp53 and its isoforms Δ133p53 and Δ160p53 on endogenous wild-type p53 aggregation, A549 cells were transfected with 2 μg of V5-tagged p53 expression plasmids, with or without etoposide (20μM for 8h) treatment. Western blot analysis was done with the anti-V5 (rabbit) to detect V5-tagged proteins and anti-DO-1 (mouse), the latter detects both endogenous wild-type p53 and V5-tagged FLp53. The merged image corresponds to the overlay between the V5 and DO1 antibody signals. (B) H1299 cells (p53-null) were transfected with 2 μg V5tagged p53 expression plasmids or the empty vector control pcDNA3.1. Western blot analysis was done with the anti-V5 (mouse) antibody.

(2) Can presence of mutant P53 isoforms can cause functional impairment of wild type full length endogenous P53? That could be tested as well using similar ChIP assay authors has performed, but instead of antibody against the Tagged protein if the authors could check endogenous P53 enrichment in the gene promoter such as P21 following overexpression of mutant isoforms. May be introducing a condition such as DNA damage in such experiment might help where endogenous P53 is induced and more prone to bind to P53 target such as P21.

Thank you very much for your valuable comments and suggestions. To investigate the potential functional impairment of endogenous wild-type p53 by p53 isoforms, we initially utilized A549 cells (p53 wild-type), aiming to monitor endogenous wild-type p53 expression following DNA damage. However, as mentioned and demonstrated in Author response image 1, endogenous p53 expression was too low to be detected under these conditions, making the ChIP assay for analyzing endogenous p53 activity unfeasible. Thus, we decided to utilize plasmid-based expression of FLp53 and focus on the potential functional impairment induced by the isoforms.

(3) On similar lines, authors described:"To test this hypothesis, we escalated the ratio of FLp53 to isoforms to 1:10. As expected, the activity of all four promoters decreased significantly at this ratio (Figure 4A-D). Notably, Δ160p53 showed a more potent inhibitory effect than Δ133p53 at the 1:5 ratio on all promoters except for the p21 promoter, where their impacts were similar (Figure 4E-H). However, at the 1:10 ratio, Δ133p53 and Δ160p53 had similar effects on all transactivation except for the MDM2 promoter (Figure 4E-H)."Again, in such assay authors used ratio 1:5 to 1:10 full length vs mutant. How authors justify this result in context (which is more relevant context) where one allele is Wild type (functional P53) and another allele is mutated (truncated, can induce aggregation). In this case one would except 1:1 ratio of full-length vs mutant protein, unless other regulation is going which induces expression of mutant isoforms more than wild type full length protein. Probably discussing on these lines might provide more physiological relevance to the observed data.

Thank you for raising this point regarding the physiological relevance of the ratios used in our study.

(1) In the revised manuscript (lines 193-195), we added in this direction that “The elevated Δ133p53 protein modulates p53 target genes such as miR‑34a and p21, facilitating cancer development(2, 3). To mimic conditions where isoforms are upregulated relative to FLp53, we increased the ratios to 1:5 and 1:10.” This approach aims to simulate scenarios where isoforms accumulate at higher levels than FLp53, which may be relevant in specific contexts, as also elaborated above.

(2) Regarding the issue of protein expression, where one allele is wild-type and the other is isoform, this assumption is not valid in most contexts. First, human cells have two copies of TPp53 gene (one from each parent). Second, the TP53 gene has two distinct promoters: the proximal promoter (P1) primarily regulates FLp53 and ∆40p53, whereas the second promoter (P2) regulates ∆133p53 and ∆160p53(4, 5). Additionally, ∆133TP53 is a p53 target gene(6, 7) and the expression of Δ133p53 and FLp53 is dynamic in response to various stimuli. Third, the expression of p53 isoforms is regulated at multiple levels, including transcriptional, post-transcriptional, translational, and post-translational processing(8). Moreover, different degradation mechanisms modify the protein level of p53 isoforms and FLp53(8). These differential regulation mechanisms are regulated by various stimuli, and therefore, the 1:1 ratio of FLp53 to ∆133p53 or ∆160p53 may be valid only under certain physiological conditions. In line with this, varied expression levels of FLp53 and its isoforms, including ∆133p53 and ∆160p53, have been reported in several studies(3, 4, 9, 10).

(3) In our study, using the pcDNA 3.1 vector under the human cytomegalovirus (CMV) promoter, we observed moderately higher expression levels of ∆133p53 and ∆160p53 relative to FLp53 (Author response image 1B). This overexpression scenario provides a model for studying conditions where isoform accumulation might surpass physiological levels, impacting FLp53 function. By employing elevated ratios of these isoforms to FLp53, we aim to investigate the potential effects of isoform accumulation on FLp53.

(4) Finally does this altered function of full length P53 (preferably endogenous one) in presence of truncated P53 has any phenotypic consequence on the cells (if authors choose a cell type which is having wild type functional P53). Doing assay such as apoptosis/cell cycle could help us to get this visualization.

Thank you for your insightful comments. In the experiment with A549 cells (p53 wild-type), endogenous p53 levels were too low to be detected, even after DNA damage induction. The evaluation of the function of endogenous p53 in the presence of isoforms is hindered, as mentioned above. In the revised manuscript, we utilized H1299 cells with overexpressed proteins for apoptosis studies using the Caspase-Glo 3/7 assay (Figure 7). This has been shown in the Results section (lines 254-269). “The Δ133p53 and Δ160p53 proteins block pro-apoptotic function of FLp53.

One of the physiological read-outs of FLp53 is its ability to induce apoptotic cell death(11). To investigate the effects of p53 isoforms Δ133p53 and Δ160p53 on FLp53-induced apoptosis, we measured caspase-3 and -7 activities in H1299 cells expressing different p53 isoforms (Figure 7). Caspase activation is a key biochemical event in apoptosis, with the activation of effector caspases (caspase-3 and -7) ultimately leading to apoptosis(12). The caspase-3 and -7 activities induced by FLp53 expression was approximately 2.5 times higher than that of the control vector (Figure 7). Co-expression of FLp53 and the isoforms Δ133p53 or Δ160p53 at a ratio of 1: 5 significantly diminished the apoptotic activity of FLp53 (Figure 7). This result aligns well with our reporter gene assay, which demonstrated that elevated expression of Δ133p53 and Δ160p53 impaired the expression of apoptosis-inducing genes BAX and PUMA (Figure 4G and H). Moreover, a reduction in the apoptotic activity of FLp53 was observed irrespective of whether Δ133p53 or Δ160p53 protein was expressed with or without a FLAG tag (Figure 7). This result, therefore, also suggests that the FLAG tag does not affect the apoptotic activity or other physiological functions of FLp53 and its isoforms. Overall, the overexpression of p53 isoforms Δ133p53 and Δ160p53 significantly attenuates FLp53-induced apoptosis, independent of the protein tagging with the FLAG antibody epitope.”

Referees cross-commentingI think the comments from the other reviewers are very much reasonable and logical.Especially all 3 reviewers have indicated, a better way to visualize the aggregation of full-length wild type P53 by truncated P53 (such as looking at endogenous P53# by reviewer 1, having fluorescent tag #by reviewer 2 and reviewer 3 raised concern on the FLAG tag) would add more value to the observation.

Thank you for these comments. The endogenous p53 protein was undetectable in A549 cells induced by etoposide (Figure R1A). Therefore, we conducted experiments using FLAG/V5-tagged FLp53. To avoid any potential side effects of the FLAG tag on p53 aggregation, we introduced untagged p53 isoforms in the H1299 cells and performed subcellular fractionation. Our revised results, consistent with previous FLAG-tagged p53 isoforms findings, demonstrate that co-expression of untagged isoforms with FLAG-tagged FLp53 significantly induced the aggregation of FLAG-FLp53, while no aggregation was observed when FLAG-tagged FLp53 was expressed alone (Supplementary Figure 6). These results clearly indicate that the FLAG tag itself does not contribute to protein aggregation.

Additionally, we utilized the A11 antibody to detect protein aggregation, providing additional validation (Figure 8 from Jean-Christophe Bourdon et al. Genes Dev. 2005;19:2122-2137). Given that the fluorescent proteins (~30 kDa) are substantially bigger than the tags used here (~1 kDa) and may influence oligomerization (especially GFP), stability, localization, and function of p53 and its isoforms, we avoided conducting these vital experiments with such artificial large fusions.

**Reviewer #1 (Significance):**
The work in significant, since it points out more mechanistic insight how wild type full length P53 could be inactivated in the presence of truncated isoforms, this might offer new opportunity to recover P53 function as treatment strategies against cancer.

Thank you for your insightful comments. We appreciate your recognition of the significance of our work in providing mechanistic insights into how wild-type FLp53 can be inactivated by truncated isoforms. We agree that these findings have potential for exploring new strategies to restore p53 function as a therapeutic approach against cancer.

**Reviewer #2 (Evidence, reproducibility and clarity):**
The manuscript by Zhao and colleagues presents a novel and compelling study on the p53 isoforms, Δ133p53 and Δ160p53, which are associated with aggressive cancer types. The main objective of the study was to understand how these isoforms exert a dominant negative effect on full-length p53 (FLp53). The authors discovered that the Δ133p53 and Δ160p53 proteins exhibit impaired binding to p53-regulated promoters. The data suggest that the predominant mechanism driving the dominant-negative effect is the coaggregation of FLp53 with Δ133p53 and Δ160p53.This study is innovative, well-executed, and supported by thorough data analysis. However, the authors should address the following points:(1) Introduction on Aggregation and Co-aggregation: Given that the focus of the study is on the aggregation and co-aggregation of the isoforms, the introduction should include a dedicated paragraph discussing this issue. There are several original research articles and reviews that could be cited to provide context.

Thank you very much for the valuable comments. We have added the following paragraph in the revised manuscript (lines 74-82): “Protein aggregation has become a central focus of modern biology research and has documented implications in various diseases, including cancer(13, 14, 15). Protein aggregates can be of different types ranging from amorphous aggregates to highly structured amyloid or fibrillar aggregates, each with different physiological implications. In the case of p53, whether protein aggregation, and in particular, co-aggregation with large N-terminal deletion isoforms, plays a mechanistic role in its inactivation is yet underexplored. Interestingly, the Δ133p53β isoform has been shown to aggregate in several human cancer cell lines(16). Additionally, the Δ40p53α isoform exhibits a high aggregation tendency in endometrial cancer cells(17). Although no direct evidence exists for Δ160p53 yet, these findings imply that p53 isoform aggregation may play a major role in their mechanisms of actions.”

(2) Antibody Use for Aggregation: To strengthen the evidence for aggregation, the authors should consider using antibodies that specifically bind to aggregates.

Thank you for your insightful suggestion. We addressed protein aggregation using the A11 antibody which specifically recognizes amyloid-like protein aggregates. We analyzed insoluble nuclear pellet samples prepared under identical conditions as described in Figure 6B. To confirm the presence of p53 proteins, we employed the anti-p53 M19 antibody (Santa Cruz, Cat No. sc-1312) to detect bands corresponding to FLp53 and its isoforms Δ133p53 and Δ160p53. The monomer FLp53 was not detected (Figure 8, lower panel, Jean-Christophe Bourdon et al. Genes Dev. 2005;19:2122-2137), which may be attributed to the lower binding affinity of the anti-p53 M19 antibody to it. These samples were also immunoprecipitated using the A11 antibody (Thermo Fischer Scientific, Cat No. AHB0052) to detect aggregated proteins. Interestingly, FLp53 and its isoforms, Δ133p53 and Δ160p53, were clearly visible with Anti-A11 antibody when co-expressed at a 1:5 ratio suggesting that they underwent co-aggregation. However, no FLp53 aggregates were observed when it was expressed alone (Author response image 2). These results support the conclusion in our manuscript that Δ133p53 and Δ160p53 drive FLp53 aggregation.

**Author response image 2. sa3fig2:** Induction of FLp53 Aggregation by p53 Isoforms Δ133p53 and Δ160p53. H1299 cells transfected with the FLAG-tagged FLp53 and V5-tagged Δ133p53 or Δ160p53 at a 1:5 ratio. The cells were subjected to subcellular fractionation, and the resulting insoluble nuclear pellet was resuspended in RIPA buffer. The samples were heated at 95°C until the pellet was completely dissolved, and then analyzed by Western blotting. Immunoprecipitation was performed using the A11 antibody, which specifically recognizes amyloid protein aggregates, and the anti-p53 M19 antibody, which detects FLp53 as well as its isoforms Δ133p53 and Δ160p53.

(3) Fluorescence Microscopy: Live-cell fluorescence microscopy could be employed to enhance visualization by labeling FLp53 and the isoforms with different fluorescent markers (e.g., EGFP and mCherry tags).

We appreciate the suggestion to use live-cell fluorescence microscopy with EGFP and mCherry tags for the visualization FLp53 and its isoforms. While we understand the advantages of live-cell imaging with EGFP / mCherry tags, we restrained us from doing such fusions as the GFP or corresponding protein tags are very big (~30 kDa) with respect to the p53 isoform variants (~30 kDa). Other studies have shown that EGFP and mCherry fusions can alter protein oligomerization, solubility and aggregation(18, 19) Moreover, most fluorescence proteins are prone to dimerization (i.e. EGFP) or form obligate tetramers (DsRed)(20, 21, 22), potentially interfering with the oligomerization and aggregation properties of p53 isoforms, particularly Δ133p53 and Δ160p53.

Instead, we utilized FLAG- or V5-tag-based immunofluorescence microscopy, a well-established and widely accepted method for visualizing p53 proteins. This method provided precise localization and reliable quantitative data, which we believe meet the needs of the current study. We believe our chosen method is both appropriate and sufficient for addressing the research question.

**Reviewer #2 (Significance):**
The manuscript by Zhao and colleagues presents a novel and compelling study on the p53 isoforms, Δ133p53 and Δ160p53, which are associated with aggressive cancer types. The main objective of the study was to understand how these isoforms exert a dominant negative effect on full-length p53 (FLp53). The authors discovered that the Δ133p53 and Δ160p53 proteins exhibit impaired binding to p53-regulated promoters. The data suggest that the predominant mechanism driving the dominant-negative effect is the coaggregation of FLp53 with Δ133p53 and Δ160p53.

We sincerely thank the reviewer for the thoughtful and positive comments on our manuscript and for highlighting the significance of our findings on the p53 isoforms, Δ133p53 and Δ160p53.

**Reviewer #3 (Evidence, reproducibility and clarity):**
In this manuscript entitled "Δ133p53 and Δ160p53 isoforms of the tumor suppressor protein p53 exert dominant-negative effect primarily by coaggregation", the authors suggest that the Δ133p53 and Δ160p53 isoforms have high aggregation propensity and that by co-aggregating with canonical p53 (FLp53), they sequestrate it away from DNA thus exerting a dominantnegative effect over it.First, the authors should make it clear throughout the manuscript, including the title, that they are investigating Δ133p53α and Δ160p53α since there are 3 Δ133p53 isoforms (α, β, γ), and 3 Δ160p53 isoforms (α, β, γ).

Thank you for your suggestion. We understand the importance of clearly specifying the isoforms under study. Following your suggestion, we have added α in the title, abstract, and introduction and added the following statement in the Introduction (lines 57-59): “For convenience and simplicity, we have written Δ133p53 and Δ160p53 to represent the α isoforms (Δ133p53α and Δ160p53α) throughout this manuscript.”

One concern is that the authors only consider and explore Δ133p53α and Δ160p53α isoforms as exclusively oncogenic and FLp53 dominant-negative while not discussing evidences of different activities. Indeed, other manuscripts have also shown that Δ133p53α is non-oncogenic and non-mutagenic, do not antagonize every single FLp53 functions and are sometimes associated with good prognosis. To cite a few examples:(1) Hofstetter G. et al. D133p53 is an independent prognostic marker in p53 mutant advanced serous ovarian cancer. Br. J. Cancer 2011, 105, 15931599.(2) Bischof, K. et al. Influence of p53 Isoform Expression on Survival in HighGrade Serous Ovarian Cancers. Sci. Rep. 2019, 9,5244.(3) Knezovi´c F. et al. The role of p53 isoforms' expression and p53 mutation status in renal cell cancer prognosis. Urol. Oncol. 2019, 37, 578.e1578.e10.(4) Gong, L. et al. p53 isoform D113p53/D133p53 promotes DNA doublestrand break repair to protect cell from death and senescence in response to DNA damage. Cell Res. 2015, 25, 351-369.(5) Gong, L. et al. p53 isoform D133p53 promotes efficiency of induced pluripotent stem cells and ensures genomic integrity during reprogramming. Sci. Rep. 2016, 6, 37281.(6) Horikawa, I. et al. D133p53 represses p53-inducible senescence genes and enhances the generation of human induced pluripotent stem cells. Cell Death Differ. 2017, 24, 1017-1028.(7) Gong, L. p53 coordinates with D133p53 isoform to promote cell survival under low-level oxidative stress. J. Mol. Cell Biol. 2016, 8, 88-90.

Thank you very much for your comment and for highlighting these important studies.

We agree that Δ133p53 isoforms exhibit complex biological functions, with both oncogenic and non-oncogenic potentials. However, our mission here was primarily to reveal the molecular mechanism for the dominant-negative effects exerted by the Δ133p53α and Δ160p53α isoforms on FLp53 for which the Δ133p53α and Δ160p53α isoforms are suitable model systems. Exploring the oncogenic potential of the isoforms is beyond the scope of the current study and we have not claimed anywhere that we are reporting that. We have carefully revised the manuscript and replaced the respective terms e.g. ‘prooncogenic activity’ with ‘dominant-negative effect’ in relevant places (e.g. line 90). We have now also added a paragraph with suitable references that introduces the oncogenic and non-oncogenic roles of the p53 isoforms.

After reviewing the papers you cited, we are not sure that they reflect on oncogenic /non-oncogenic role of the Δ133p53α isoform in different cancer cases. Although our study is not about the oncogenic potential of the isoforms, we have summarized the key findings below:

(1) Hofstetter et al., 2011: Demonstrated that Δ133p53α expression improved recurrence-free and overall survival in a p53 mutant induced advanced serous ovarian cancer, suggesting a potential protective role in this context.

(2) Bischof et al., 2019: Found that Δ133p53 mRNA can improve overall survival in high-grade serous ovarian cancers. However, out of 31 patients, only 5 belong to the TP53 wild-type group, while the others carry TP53 mutations.

(3) Knezović et al., 2019: Reported downregulation of Δ133p53 in renal cell carcinoma tissues with wild-type p53 compared to normal adjacent tissue, indicating a potential non-oncogenic role, but not conclusively demonstrating it.

(4) Gong et al., 2015: Showed that Δ133p53 antagonizes p53-mediated apoptosis and promotes DNA double-strand break repair by upregulating RAD51, LIG4, and RAD52 independently of FLp53.

(5) Gong et al., 2016: Demonstrated that overexpression of Δ133p53 promotes efficiency of cell reprogramming by its anti-apoptotic function and promoting DNA DSB repair. The authors hypotheses that this mechanism is involved in increasing RAD51 foci formation and decrease γH2AX foci formation and chromosome aberrations in induced pluripotent stem (iPS) cells, independent of FL p53.

(6) Horikawa et al., 2017: Indicated that induced pluripotent stem cells derived from fibroblasts that overexpress Δ133p53 formed noncancerous tumors in mice compared to induced pluripotent stem cells derived from fibroblasts with complete p53 inhibition. Thus, Δ133p53 overexpression is "non- or less oncogenic and mutagenic" compared to complete p53 inhibition, but it still compromises certain p53-mediated tumor-suppressing pathways. “Overexpressed Δ133p53 prevented FL-p53 from binding to the regulatory regions of p21WAF1 and miR-34a promoters, providing a mechanistic basis for its dominant-negative

inhibition of a subset of p53 target genes.”

(7) Gong, 2016: Suggested that Δ133p53 promotes cell survival under lowlevel oxidative stress, but its role under different stress conditions remains uncertain.

We have revised the Introduction to provide a more balanced discussion of Δ133p53’s dule role (lines 62-73):

“The Δ133p53 isoform exhibit complex biological functions, with both oncogenic and non-oncogenic potentials. Recent studies demonstrate the non-oncogenic yet context-dependent role of the Δ133p53 isoform in cancer development. Δ133p53 expression has been reported to correlate with improved survival in patients with TP53 mutations(23, 24), where it promotes cell survival in a nononcogenic manner(25, 26), especially under low oxidative stress(27). Alternatively, other recent evidences emphasize the notable oncogenic functions of Δ133p53 as it can inhibit p53-dependent apoptosis by directly interacting with the FLp53 (4, 6). The oncogenic function of the newly identified Δ160p53 isoform is less known, although it is associated with p53 mutation-driven tumorigenesis(28) and in melanoma cells’ aggressiveness(10). Whether or not the Δ160p53 isoform also impedes FLp53 function in a similar way as Δ133p53 is an open question. However, these p53 isoforms can certainly compromise p53-mediated tumor suppression by interfering with FLp53 binding to target genes such as p21 and miR-34a(2, 29) by dominant-negative effect, the exact mechanism is not known.” On the figures presented in this manuscript, I have three major concerns:

(1) Most results in the manuscript rely on the overexpression of the FLAGtagged or V5-tagged isoforms. The validation of these construct entirely depends on Supplementary figure 3 which the authors claim "rules out the possibility that the FLAG epitope might contribute to this aggregation. However, I am not entirely convinced by that conclusion. Indeed, the ratio between the "regular" isoform and the aggregates is much higher in the FLAG-tagged constructs than in the V5-tagged constructs. We can visualize the aggregates easily in the FLAG-tagged experiment, but the imaging clearly had to be overexposed (given the white coloring demonstrating saturation of the main bands) to visualize them in the V5-tagged experiments. Therefore, I am not convinced that an effect of the FLAG-tag can be ruled out and more convincing data should be added.

Thank you for raising this important concern. We have carefully considered your comments and have made several revisions to clarify and strengthen our conclusions.

First, to address the potential influence of the FLAG and V5 tags on p53 isoform aggregation, we have revised Figure 2 and removed the previous Supplementary Figure 3, where non-specific antibody bindings and higher molecular weight aggregates were not clearly interpretable. In the revised Figure 2, we have removed these potential aggregates, improving the clarity and accuracy of the data.

To further rule out any tag-related artifacts, we conducted a coimmunoprecipitation assay with FLAG-tagged FLp53 and untagged Δ133p53 and Δ160p53 isoforms. The results (now shown in the new Supplementary Figure 3) completely agree with our previous result with FLAG-tagged and V5tagged Δ133p53 and Δ160p53 isoforms and show interaction between the partners. This indicates that the FLAG / V5-tags do not influence / interfere with the interaction between FLp53 and the isoforms. We have still used FLAGtagged FLp53 as the endogenous p53 was undetectable and the FLAG-tagged FLp53 did not aggregate alone.

In the revised paper, we added the following sentences (Lines 146-152): “To rule out the possibility that the observed interactions between FLp53 and its isoforms Δ133p53 and Δ160p53 were artifacts caused by the FLAG and V5 antibody epitope tags, we co-expressed FLAG-tagged FLp53 with untagged Δ133p53 and Δ160p53. Immunoprecipitation assays demonstrated that FLAGtagged FLp53 could indeed interact with the untagged Δ133p53 and Δ160p53 isoforms (Supplementary Figure 3, lanes 3 and 4), confirming formation of hetero-oligomers between FLp53 and its isoforms. These findings demonstrate that Δ133p53 and Δ160p53 can oligomerize with FLp53 and with each other.”

Additionally, we performed subcellular fractionation experiments to compare the aggregation and localization of FLAG-tagged FLp53 when co-expressed either with V5-tagged or untagged Δ133p53/Δ160p53. In these experiments, the untagged isoforms also induced FLp53 aggregation, mirroring our previous results with the tagged isoforms (Supplementary Figure 5). We’ve added this result in the revised manuscript (lines 236-245): “To exclude the possibility that FLAG or V5 tags contribute to protein aggregation, we also conducted subcellular fractionation of H1299 cells expressing FLAG-tagged FLp53 along with untagged Δ133p53 or Δ160p53 at a 1:5 ratio. The results showed (Supplementary Figure 6) a similar distribution of FLp53 across cytoplasmic, nuclear, and insoluble nuclear fractions as in the case of tagged Δ133p53 or Δ160p53 (Figure 6A to D). Notably, the aggregation of untagged Δ133p53 or Δ160p53 markedly promoted the aggregation of FLAG-tagged FLp53 (Supplementary Figure 6B and D), demonstrating that the antibody epitope tags themselves do not contribute to protein aggregation.”

We’ve also discussed this in the Discussion section (lines 349-356): “In our study, we primarily utilized an overexpression strategy involving FLAG/V5tagged proteins to investigate the effects of p53 isoforms Δ133p53 and Δ160p53 on the function of FLp53. To address concerns regarding potential overexpression artifacts, we performed the co-immunoprecipitation (Supplementary Figure 6) and caspase-3 and -7 activity (Figure 7) experiments with untagged Δ133p53 and Δ160p53. In both experimental systems, the untagged proteins behaved very similarly to the FLAG/V5 antibody epitopecontaining proteins (Figures 6 and 7 and Supplementary Figure 6). Hence, the C-terminal tagging of FLp53 or its isoforms does not alter the biochemical and physiological functions of these proteins.”

In summary, the revised data set and newly added experiments provide strong evidence that neither the FLAG nor the V5 tag contributes to the observed p53 isoform aggregation.

(2) The authors demonstrate that to visualize the dominant-negative effect, Δ133p53α and Δ160p53α must be "present in a higher proportion than FLp53 in the tetramer" and the need at least a transfection ratio 1:5 since the 1:1 ration shows no effect. However, in almost every single cell type, FLp53 is far more expressed than the isoforms which make it very unlikely to reach such stoichiometry in physiological conditions and make me wonder if this mechanism naturally occurs at endogenous level. This limitation should be at least discussed.

Thank you for your insightful comment. However, evidence suggests that the expression levels of these isoforms such as Δ133p53, can be significantly elevated relative to FLp53 in certain physiological conditions(3, 4, 9). For example, in some breast tumors, with Δ133p53 mRNA is expressed at a much levels than FLp53, suggesting a distinct expression profile of p53 isoforms compared to normal breast tissue(4). Similarly, in non-small cell lung cancer and the A549 lung cancer cell line, the expression level of Δ133p53 transcript is significantly elevated compared to non-cancerous cells(3). Moreover, in specific cholangiocarcinoma cell lines, the Δ133p53 /TAp53 expression ratio has been reported to increase to as high as 3:1(9). These observations indicate that the dominant-negative effect of isoform Δ133p53 on FLp53 can occur under certain pathological conditions where the relative amounts of the FLp53 and the isoforms would largely vary. Since data on the Δ160p53 isoform are scarce, we infer that the long N-terminal truncated isoforms may share a similar mechanism.

(3) Figure 5C: I am concerned by the subcellular location of the Δ133p53α and Δ160p53α as they are commonly considered nuclear and not cytoplasmic as shown here, particularly since they retain the 3 nuclear localization sequences like the FLp53 (Bourdon JC et al. 2005; Mondal A et al. 2018; Horikawa I et al, 2017; Joruiz S. et al, 2024). However, Δ133p53α can form cytoplasmic speckles (Horikawa I et al, 2017) when it colocalizes with autophagy markers for its degradation.The authors should discuss this issue. Could this discrepancy be due to the high overexpression level of these isoforms? A co-staining with autophagy markers (p62, LC3B) would rule out (or confirm) activation of autophagy due to the overwhelming expression of the isoform.

Thank you for your thoughtful comments. We have thoroughly reviewed all the papers you recommended (Bourdon JC et al., 2005; Mondal A et al., 2018; Horikawa I et al., 2017; Joruiz S. et al., 2024)(4, 29, 30, 31). Among these, only the study by Bourdon JC et al. (2005) provided data regarding the localization of Δ133p53(4). Interestingly, their findings align with our observations, indicating that the protein does not exhibit predominantly nuclear localization in the Figure 8 from Jean-Christophe Bourdon et al. Genes Dev. 2005;19:2122-2137. The discrepancy may be caused by a potentially confusing statement in that paper(4).

The localization of p53 is governed by multiple factors, including its nuclear import and export(32). The isoforms Δ133p53 and Δ160p53 contain three nuclear localization sequences (NLS)(4). However, the isoforms Δ133p53 and Δ160p53 were potentially trapped in the cytoplasm by aggregation and masking the NLS. This mechanism would prevent nuclear import.

Further, we acknowledge that Δ133p53 co-aggregates with autophagy substrate p62/SQSTM1 and autophagosome component LC3B in cytoplasm by autophagic degradation during replicative senescence(33). We agree that high overexpression of these aggregation-prone proteins may induce endoplasmic reticulum (ER) stress and activates autophagy(34). This could explain the cytoplasmic localization in our experiments. However, it is also critical to consider that we observed aggregates in both the cytoplasm and the nucleus (Figures 6B and E and Supplementary Figure 6B). While cytoplasmic localization may involve autophagy-related mechanisms, the nuclear aggregates likely arise from intrinsic isoform properties, such as altered protein folding, independent of autophagy. These dual localizations reflect the complex behavior of Δ133p53 and Δ160p53 isoforms under our experimental conditions.

In the revised manuscript, we discussed this in Discussion (lines 328-335): “Moreover, the observed cytoplasmic isoform aggregates may reflect autophagy-related degradation, as suggested by the co-localization of Δ133p53 with autophagy substrate p62/SQSTM1 and autophagosome component LC3B(33). High overexpression of these aggregation-prone proteins could induce endoplasmic reticulum stress and activate autophagy(34). Interestingly, we also observed nuclear aggregation of these isoforms (Figure 6B and E and Supplementary Figure 6B), suggesting that distinct mechanisms, such as intrinsic properties of the isoforms, may govern their localization and behavior within the nucleus. This dual localization underscores the complexity of Δ133p53 and Δ160p53 behavior in cellular systems.”

Minor concerns:- Figure 1A: the initiation of the "Δ140p53" is shown instead of "Δ40p53"

Thank you! The revised Figure 1A has been created in the revised paper.

- Figure 2A: I would like to see the images cropped a bit higher, so the cut does not happen just above the aggregate bands

Thank you for this suggestion. We’ve changed the image and the new Figure 2 has been shown in the revised paper.

- Figure 3C: what ratio of FLp53/Delta isoform was used?

We have added the ratio in the figure legend of Figure 3C (lines 845-846) “Relative DNA-binding of the FLp53-FLAG protein to the p53-target gene promoters in the presence of the V5-tagged protein Δ133p53 or Δ160p53 at a 1: 1 ratio.”

- Figure 3C suggests that the "dominant-negative" effect is mostly senescencespecific as it does not affect apoptosis target genes, which is consistent with Horikawa et al, 2017 and Gong et al, 2016 cited above. Furthermore, since these two references and the others from Gong et al. show that Δ133p53α increases DNA repair genes, it would be interesting to look at RAD51, RAD52 or Lig4, and maybe also induce stress.

Thank you for your thoughtful comments and suggestions. In Figure 3C, the presence of Δ133p53 or Δ160p53 only significantly reduced the binding of FLp53 to the p21 promoter. However, isoforms Δ133p53 and Δ160p53 demonstrated a significant loss of DNA-binding activity at all four promoters: p21, MDM2, and apoptosis target genes BAX and PUMA (Figure 3B). This result suggests that Δ133p53 and Δ160p53 have the potential to influence FLp53 function due to their ability to form hetero-oligomers with FLp53 or their intrinsic tendency to aggregate. To further investigate this, we increased the isoform to FLp53 ratio in Figure 4, which demonstrate that the isoforms Δ133p53 and Δ160p53 exert dominant-negative effects on the function of FLp53.

These results demonstrate that the isoforms can compromise p53-mediated pathways, consistent with Horikawa et al. (2017), which showed that Δ133p53α overexpression is "non- or less oncogenic and mutagenic" compared to complete p53 inhibition, but still affects specific tumor-suppressing pathways. Furthermore, as noted by Gong et al. (2016), Δ133p53’s anti-apoptotic function under certain conditions is independent of FLp53 and unrelated to its dominantnegative effects.

We appreciate your suggestion to investigate DNA repair genes such as RAD51, RAD52, or Lig4, especially under stress conditions. While these targets are intriguing and relevant, we believe that our current investigation of p53 targets in this manuscript sufficiently supports our conclusions regarding the dominant-negative effect. Further exploration of additional p53 target genes, including those involved in DNA repair, will be an important focus of our future studies.

- Figure 5A and B: directly comparing the level of FLp53 expressed in cytoplasm or nucleus to the level of Δ133p53α and Δ160p53α expressed in cytoplasm or nucleus does not mean much since these are overexpressed proteins and therefore depend on the level of expression. The authors should rather compare the ratio of cytoplasmic/nuclear FLp53 to the ratio of cytoplasmic/nuclear Δ133p53α and Δ160p53α.

Thank you very much for this valuable suggestion. In the revised paper, Figure 5B has been recreated. Changes have been made in lines 214215: “The cytoplasm-to-nucleus ratio of Δ133p53 and Δ160p53 was approximately 1.5-fold higher than that of FLp53 (Figure 5B).”

Referees cross-commentingI agree that the system needs to be improved to be more physiological.Just to precise, the D133 and D160 isoforms are not truncated mutants, they are naturally occurring isoforms expressed in almost every normal human cell type from an internal promoter within the TP53 gene.Using overexpression always raises concerns, but in this case, I am even more careful because the isoforms are almost always less expressed than the FLp53, and here they have to push it 5 to 10 times more expressed than the FLp53 to see the effect which make me fear an artifact effect due to the overwhelming overexpression (which even seems to change the normal localization of the protein).To visualize the endogenous proteins, they will have to change cell line as the H1299 they used are p53 null.

Thank you for these comments. We’ve addressed the motivation of overexpression in the above responses. We needed to use the plasmid constructs in the p53-null cells to detect the proteins but the expression level was certainly not ‘overwhelmingly high’.

First, we tried the A549 cells (p53 wild-type) under DNA damage conditions, but the endogenous p53 protein was undetectable. Second, several studies reported increased Δ133p53 level compared to wild-type p53 and that it has implications in tumor development(2, 3, 4, 9). Third, the apoptosis activity of H1299 cells overexpressing p53 proteins was analyzed in the revised manuscript (Figure 7). The apoptotic activity induced by FLp53 expression was approximately 2.5 times higher than that of the control vector under identical plasmid DNA transfection conditions (Figure 7). These results rule out the possibility that the plasmid-based expression of p53 and its isoforms introduced artifacts in the results. We’ve discussed this in the Results section (lines 254269).

**Reviewer #3 (Significance):**
Overall, the paper is interesting particularly considering the range of techniques used which is the main strength.The main limitation to me is the lack of contradictory discussion as all argumentation presents Δ133p53α and Δ160p53α exclusively as oncogenic and strictly FLp53 dominant-negative when, particularly for Δ133p53α, a quite extensive literature suggests a not so clear-cut activity.The aggregation mechanism is reported for the first time for Δ133p53α and Δ160p53α, although it was already published for Δ40p53α, Δ133p53β or in mutant p53.This manuscript would be a good basic research addition to the p53 field to provide insight in the mechanism for some activities of some p53 isoforms.My field of expertise is the p53 isoforms which I have been working on for 11 years in cancer and neuro-degenerative diseases

Thank you very much for your positive and critical comments. We’ve included a fair discussion on the oncogenic and non-oncogenic function of Δ133p53 in the Introduction following your suggestion (lines 62-73).

References

(1) Pitolli C, Wang Y, Candi E, Shi Y, Melino G, Amelio I. p53-Mediated Tumor Suppression: DNA-Damage Response and Alternative Mechanisms. Cancers 11, (2019).

(2) Fujita K, et al. p53 isoforms Delta133p53 and p53beta are endogenous regulators of replicative cellular senescence. Nature cell biology 11, 1135-1142 (2009).

(3) Fragou A, et al. Increased Δ133p53 mRNA in lung carcinoma corresponds with reduction of p21 expression. Molecular medicine reports 15, 1455-1460 (2017).

(4) Bourdon JC, et al. p53 isoforms can regulate p53 transcriptional activity. Genes & development 19, 2122-2137 (2005).

(5) Ghosh A, Stewart D, Matlashewski G. Regulation of human p53 activity and cell localization by alternative splicing. Molecular and cellular biology 24, 7987-7997 (2004).

(6) Aoubala M, et al. p53 directly transactivates Δ133p53α, regulating cell fate outcome in response to DNA damage. Cell death and differentiation 18, 248-258 (2011).

(7) Marcel V, et al. p53 regulates the transcription of its Delta133p53 isoform through specific response elements contained within the TP53 P2 internal promoter. Oncogene 29, 2691-2700 (2010).

(8) Zhao L, Sanyal S. p53 Isoforms as Cancer Biomarkers and Therapeutic Targets. Cancers 14, (2022).

(9) Nutthasirikul N, Limpaiboon T, Leelayuwat C, Patrakitkomjorn S, Jearanaikoon P. Ratio disruption of the ∆133p53 and TAp53 isoform equilibrium correlates with poor clinical outcome in intrahepatic cholangiocarcinoma. International journal of oncology 42, 1181-1188 (2013).

(10) Tadijan A, et al. Altered Expression of Shorter p53 Family Isoforms Can Impact Melanoma Aggressiveness. Cancers 13, (2021).

(11) Aubrey BJ, Kelly GL, Janic A, Herold MJ, Strasser A. How does p53 induce apoptosis and how does this relate to p53-mediated tumour suppression? Cell death and differentiation 25, 104-113 (2018).

(12) Ghorbani N, Yaghubi R, Davoodi J, Pahlavan S. How does caspases regulation play role in cell decisions? apoptosis and beyond. Molecular and cellular biochemistry 479, 1599-1613 (2024).

(13) Petronilho EC, et al. Oncogenic p53 triggers amyloid aggregation of p63 and p73 liquid droplets. Communications chemistry 7, 207 (2024).

(14) Forget KJ, Tremblay G, Roucou X. p53 Aggregates penetrate cells and induce the coaggregation of intracellular p53. PloS one 8, e69242 (2013).

(15) Farmer KM, Ghag G, Puangmalai N, Montalbano M, Bhatt N, Kayed R. P53 aggregation, interactions with tau, and impaired DNA damage response in Alzheimer's disease. Acta neuropathologica communications 8, 132 (2020).

(16) Arsic N, et al. Δ133p53β isoform pro-invasive activity is regulated through an aggregation-dependent mechanism in cancer cells. Nature communications 12, 5463 (2021).

(17) Melo Dos Santos N, et al. Loss of the p53 transactivation domain results in high amyloid aggregation of the Δ40p53 isoform in endometrial carcinoma cells. The Journal of biological chemistry 294, 9430-9439 (2019).

(18) Mestrom L, et al. Artificial Fusion of mCherry Enhances Trehalose Transferase Solubility and Stability. Applied and environmental microbiology 85, (2019).

(19) Kaba SA, Nene V, Musoke AJ, Vlak JM, van Oers MM. Fusion to green fluorescent protein improves expression levels of Theileria parva sporozoite surface antigen p67 in insect cells. Parasitology 125, 497-505 (2002).

(20) Snapp EL, et al. Formation of stacked ER cisternae by low affinity protein interactions. The Journal of cell biology 163, 257-269 (2003).

(21) Jain RK, Joyce PB, Molinete M, Halban PA, Gorr SU. Oligomerization of green fluorescent protein in the secretory pathway of endocrine cells. The Biochemical journal 360, 645-649 (2001).

(22) Campbell RE, et al. A monomeric red fluorescent protein. Proceedings of the National Academy of Sciences of the United States of America 99, 7877-7882 (2002).

(23) Hofstetter G, et al. Δ133p53 is an independent prognostic marker in p53 mutant advanced serous ovarian cancer. British journal of cancer 105, 1593-1599 (2011).

(24) Bischof K, et al. Influence of p53 Isoform Expression on Survival in High-Grade Serous Ovarian Cancers. Scientific reports 9, 5244 (2019).

(25) Gong L, et al. p53 isoform Δ113p53/Δ133p53 promotes DNA double-strand break repair to protect cell from death and senescence in response to DNA damage. Cell research 25, 351-369 (2015).

(26) Gong L, et al. p53 isoform Δ133p53 promotes efficiency of induced pluripotent stem cells and ensures genomic integrity during reprogramming. Scientific reports 6, 37281 (2016).

(27) Gong L, Pan X, Yuan ZM, Peng J, Chen J. p53 coordinates with Δ133p53 isoform to promote cell survival under low-level oxidative stress. Journal of molecular cell biology 8, 88-90 (2016).

(28) Candeias MM, Hagiwara M, Matsuda M. Cancer-specific mutations in p53 induce the translation of Δ160p53 promoting tumorigenesis. EMBO reports 17, 1542-1551 (2016).

(29) Horikawa I, et al. Δ133p53 represses p53-inducible senescence genes and enhances the generation of human induced pluripotent stem cells. Cell death and differentiation 24, 1017-1028 (2017).

(30) Mondal AM, et al. Δ133p53α, a natural p53 isoform, contributes to conditional reprogramming and long-term proliferation of primary epithelial cells. Cell death & disease 9, 750 (2018).

(31) Joruiz SM, Von Muhlinen N, Horikawa I, Gilbert MR, Harris CC. Distinct functions of wild-type and R273H mutant Δ133p53α differentially regulate glioblastoma aggressiveness and therapy-induced senescence. Cell death & disease 15, 454 (2024).

(32) O'Brate A, Giannakakou P. The importance of p53 location: nuclear or cytoplasmic zip code? Drug resistance updates : reviews and commentaries in antimicrobial and anticancer chemotherapy 6, 313-322 (2003).

(33) Horikawa I, et al. Autophagic degradation of the inhibitory p53 isoform Δ133p53α as a regulatory mechanism for p53-mediated senescence. Nature communications 5, 4706 (2014).

(34) Lee H, et al. IRE1 plays an essential role in ER stress-mediated aggregation of mutant huntingtin via the inhibition of autophagy flux. Human molecular genetics 21, 101-114 (2012).